# GENERATIVE AUTO-BIDDING IN LARGE-SCALE AUCTIONS VIA DIFFUSION COMPLETER-ALIGNER

## ABSTRACT

Bid optimization strategy in auto-bidding is central to computational advertising, achieving notable commercial success by optimizing advertisers' bids within constraints. Recently, generative models have revolutionizing auto-bidding by directly learning a policy from large-scale datasets. Among them, the diffuser is superior in tackling sparse-reward challenges, along with its trajectory stitching and explainability capabilities, making it well-suited for industrial auto-bidding. However, its performance could be limited by generation uncertainty, particularly regarding generations' dynamic illegitimacy and preference misalignment, which can lead to suboptimal bids and further cause poor performance when competing with other advertisers in highly competitive auctions. To address it, we propose a **C**ausal auto-**B**idding method based on a **D**iffusion completer-aligner framework, termed **CBD**. Firstly, we conduct a theoretical analysis and propose a completer to augment the training process with an extra random variable $t$ for enhancing the dynamic legitimacy between adjacent states. Then, we employ a trajectory-level return model as an aligner to refine the generated trajectories in inference, aligning more closely with advertisers' objectives. Experiments across diverse settings demonstrate that our approach not only achieves superior performance on large-scale auto-bidding benchmarks, such as a 29.9% improvement of conversion value in the challenging sparse-reward setting, but also delivers significant improvements on an online advertising platform, including a 2.0% increase in target cost.

## 1 INTRODUCTION

The rapid digital transformation of commerce has significantly expanded the reach of online ad platforms, making auto-bidding essential by automatically determining bid prices on behalf of advertisers in large-scale auctions (Ren et al., 2017). Bid optimization strategy in auto-bidding is to maximize achieved impression value for an advertiser while adhering to some economic constraints, such as cost-per-conversion (CPC), throughout an entire bidding period (Guo et al., 2024). Previous auto-bidding strategies have evolved from simple rule-based methods (Zhu et al., 2017; Yang et al., 2019) to the advanced methods based on reinforcement learning (RL) (He et al., 2021; Cai et al., 2017; Sutton, 2018), continuously pushing the boundaries of performance. With the vast amount of bidding logs available from the online system, it is getting more promising to leverage generative models to revolutionize auto-bidding by directly learning a strategy from a large-scale offline dataset that can handle the complex advertising environment while adapting to advertisers' diverse preferences.

In existing literature, two major paradigms employ generative models for decision-making tasks based on offline datasets: Decision Transformer (DT) (Chen et al., 2021; Liu et al., 2023) and Diffuser (Ajay et al., 2023; Janner et al., 2022a). DT learns to autoregressively generate bid actions upon receiving the return-to-go and observed states and actions, which relies on the single-step reward design. However, this approach is not suitable for auto-bidding because conversions, *i.e.*, the results of bidding actions, are typically very sparse in online ad environments such as in-app purchases (IAP) (Guo et al., 2024; Su et al., 2024), making it difficult to design an accurate single-step reward (Zhu et al., 2023). Diffuser offers a promising alternative by generating a trajectory of states based on accumulated rewards over multiple steps as a condition, then actions are executed according to the generation (Ajay et al., 2023). This accumulation approach significantly alleviates the single-step sparse reward issue, making it more suitable for realistic auto-bidding. Moreover, the diffuser has advantages in trajectory stitching ability and explainability (Ajay et al., 2023). DT and Diffuser were

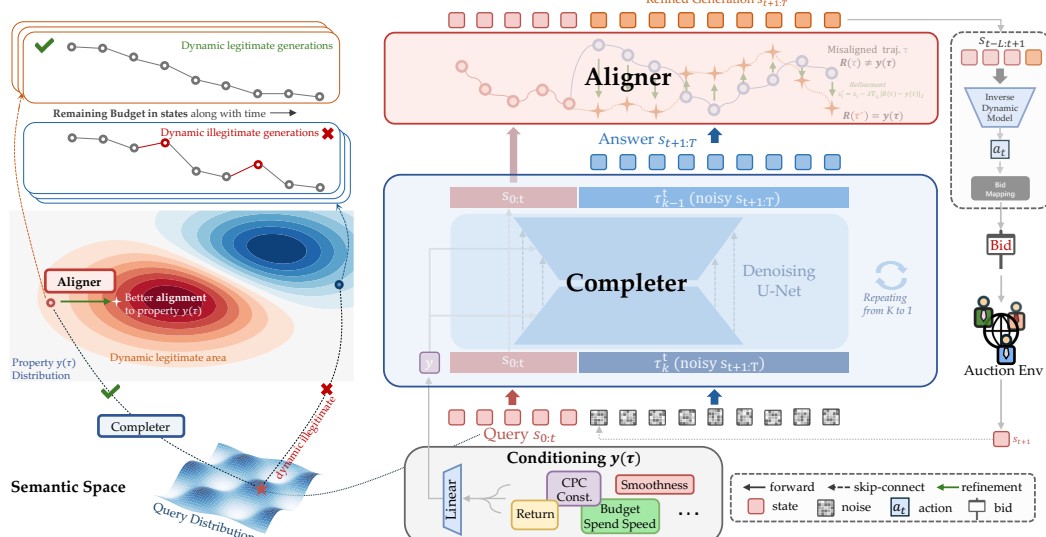

Figure 1: Overview of our CBD method: At each timestep $t$, a completer first receives a "Query" (observed states $s_{0:t}$ reflecting ad impression features and auction status) and outputs an "Answer" (generated future states $s_{t+1:T}$) within the dynamic legitimate area; The aligner then refines the generation to achieve closer alignment with the expected trajectory property; Finally, a bid is determined based on the refined trajectory and sent to the auction to compete with other advertisers.

initially developed to address offline RL tasks, aiming to stitch suboptimal sub-trajectories in offline datasets into an optimal trajectory. DT achieves this through dynamic programming, *i.e.*, generating actions and interacting with the environment to get the next state. However, this method has proven inefficient due to error accumulation in long-term decision-making compared to Diffuser, which, on the other hand, can generate an optimal trajectory by stitching via its powerful policy representation ability. Additionally, Diffuser is more explainable by revealing its planning process first and then executing strictly according to it, while DT operates as a black-box by directly outputting actions. Explainability is crucial for industrial auto-bidding services, as advertisers may discontinue the service if they observe abnormal and unexplainable behavior from the auto-bidding model.

However, we found that directly applying diffusers could result in limited performance in large-scale auctions with numerous advertisers and millions of impression opportunities (Su et al., 2024). This arises from the inherent uncertainty issue in generative models, which can occur in models ranging from simple generative models like VAEs (Kingma and Welling, 2014) to the most advanced LLMs based on auto-regressive transformers, where this phenomenon is specifically referred to as *hallucination* (Huang et al., 2025). Specifically, the generated next state may not be achievable from the current state, leading to a dynamic legitimacy issue (Ni et al., 2023). Additionally, as it is hard to train a perfect model covering all modes, the generations of diffusers can sometimes be misaligned with the advertiser's preferences (Li et al., 2024). The situation worsens in large-scale auctions, where the tolerance for bids is minimal due to high competitiveness among advertisers. Suboptimal bids caused by these issues can lead to significant loss of impression opportunities or budget waste. Previous works have tried to address these issues in offline RL tasks, typically involving single-agent, non-competitive tasks like locomotion control (Fu et al., 2020). AdaptDiffuser (Liang et al., 2023) and MetaDiffuser (Ni et al., 2023) use a forward dynamic model (FDM) for enhancing legitimacy, but a poor base model could make FDM useless (Ajay et al., 2023). DiffuserLite (Dong et al., 2024a) revises the training process as generative interpolation by training multiple diffusers, which is inefficient and could limit overall planning. Adopting causal network architectures for training a diffuser, such as DiT (Peebles and Xie, 2023) with causal attention (Yang et al., 2021), is closely related to addressing this issue, but also raises concerns about accumulated errors.

To alleviate the generation uncertainty issue of diffusers in auto-bidding for large-scale competitive auctions, we conduct an in-depth analysis of the diffuser's training and inference processes within the context of auto-bidding. Our theoretical findings indicate that directly applying diffusers can lead to unnecessary distribution mismatch errors, as defined in Eq. 21, due to a discrepancy between

the inputs used during the training and inference stages. Therefore, we begin by augmenting the denoising process during the training stage by introducing an additional random variable $t$. This reformulates model training as learning a completer, thereby mitigating the discrepancy that could lead to illegitimacy. Drawing an analogy to LLM, this process involves observing random $t$-length historical sequences as a "query" and then completing the remaining sequence as an "answer", thus enhancing the dynamic legitimacy between randomly adjacent states. However, due to the model's limited capabilities and the constantly evolving preferences of advertisers, the generated trajectory may still be suboptimal, leading to potential misalignment issues. Thus, during inference, we utilize a trajectory-level return model as an efficient aligner to adaptively refine the final generated trajectories, ensuring they align more closely with advertisers' preferences. This approach also highlights the flexibility and explainability of using a planning-based method with diffusers. Combining them leads to our **C**ausal auto-**B**idding method based on a **D**iffusion completer-aligner, termed **CBD**. To summarize, our contributions are as follows:

- We highlight the necessity of adopting diffusers in the context of industrial auto-bidding and theoretically analyze the factors hindering their development.

- We propose a novel CBD method to improve the diffuser's auto-bidding performance by alleviating the impact of generation uncertainty in both training and inference.

- Experiments show that our method excels in large-scale auto-bidding benchmarks, like achieving a notable 29.9% improvement in the challenging sparse-reward scenario similar to industrial reality. Given the sequential decision-making nature of our method, it could also provide insights and advantages for general offline RL tasks. More importantly, we successfully deploy our CBD method to an online advertising platform and achieve a significant improvement, *e.g.*, a 2.0% increase in the target cost.

## 2 PRELIMINARY AND RELATED WORK

### 2.1 PROBLEM STATEMENT OF AUTO-BIDDING

Consider a scenario where there are $H$ impression opportunities arriving sequentially, each labeled by an index $i$. An advertiser wins an impression if its bid $b_i$ surpasses those advertisers and incurs a cost $c_i$. The goal is to maximize the total value of the impressions won, represented by $\sum_i o_i v_i$, where $v_i$ denotes the impression value and $o_i$ is a binary variable indicating whether the advertiser wins impression $i$. Additionally, it is essential to consider the budget and various economic constraints, such as limiting the unit cost of specific advertising events like CPC and CPA (He et al., 2021). Therefore, under the budget constraint, the objective of auto-bidding is:

$$\text{maximize} \quad \sum_i o_i v_i \quad \text{s.t.} \quad \frac{\sum_i o_i c_{ij}}{\sum_i o_i p_{ij}} \leq C_j, \forall j. \tag{1}$$

where $p_{ij}$ is the performance indicator to $j$-th constraint provided by the advertiser and $C_j$ is the upper bound. A previous study (He et al., 2021) has shown the optimal solution:

$$b_i^* = \lambda_0 v_i + \sum_{j=1}^{J} \lambda_j p_{ij} C_j, \tag{2}$$

where $b_i^*$ is the optimal bid for impression $i$. The parameters $\lambda_j$ represent the optimal bidding parameters. Please note that in industrial ad platforms, **auto-bidding methods focus on adjusting bidding parameters at a low frequency, such as every 30 minutes, rather than for every single impression that arises in milliseconds.** Therefore, there is ample time for an advanced auto-bidding method to respond to bidding parameter adjustment requirements, even for LLMs. Appendix A also provides a detailed literature review that highlights the evolution of auto-bidding methods.

### 2.2 AUTO-BIDDING AS DECISION-MAKING

We consider a sequential decision-making framework where an auto-bidding agent interacts with the advertising environment $\mathcal{E}$ at discrete timesteps. At each timestep $t$, the agent receives a state $s_t \in \mathcal{S}$ that describes the ad impression features and auction status, and then outputs an action $a_t \in \mathcal{A}$. The action is the adjustment to the bidding parameters $\lambda_j$, *i.e.*, $a_t := \{a_t^{\lambda_0}, ..., a_t^{\lambda_J}\}$, which is then mapped to the final bid using Eq. (2). After transitioning to the next state, the env $\mathcal{E}$ emits a reward $r_t$, typically the acquired conversion value. This process repeats until the end of a bidding period, such as one day, which could be seen as a fixed-length decision-making. A detailed description of these elements can be seen in Appendix C.

**Related Work.** As the complexity of online ad environments increased, RL became a major approach for auto-bidding. USCB (He et al., 2021) and SORL (Mou et al., 2022) employ RL to maximize the value under multiple constraints. Recently, DiffBid (Guo et al., 2024), a pioneering generative auto-bidding method, directly utilizes a decision diffuser to create a planning and control scheme, but not tested in large-scale auctions. GAS (Li et al., 2024) and GAVE (Gao et al., 2025a) propose data augmentation techniques to enhance the quality of offline datasets, but they still rely on the single-step reward and could introduce instabilities from the extra value estimation (Ajay et al., 2023). Therefore, this paper focuses on exploring the potential of diffusers in large-scale auctions for contributing a superior generative backbone.

## 2.3 DIFFUSER FOR AUTO-BIDDING

The diffuser is developed for decision-making based on a diffusion model (Janner et al., 2022b; Ajay et al., 2023). For simplicity, we introduce basic elements of the decision diffuser (DD) employed by DiffBid, which aims to maximize likelihood estimation of a trajectory $x_0(\tau)$ given a condition $y(\tau)$:

$$\max_\theta \mathbb{E}_{\tau \sim \mathcal{D}}[\log p_\theta(x_0(\tau)|y(\tau))], \tag{3}$$

where $\tau$ is the trajectory index and $y(\tau)$ can be the property of the trajectory $x_0(\tau)$, such as the accumulated reward of the trajectory. For brevity, we index the states from $0$ to $T$, *i.e.*, $x_0 = \{s_0, s_1, ..., s_T\}$. Specifically, DD is built with a predefined forward noising process $q(x_{k+1}(\tau)|x_k(\tau)) := \mathcal{N}(x_{k+1}(\tau)|\sqrt{\alpha_k}x_k(\tau), (1-\alpha_k)\mathbf{I})$ and a trainable reverse denoising process $p_\theta(x_{k-1}(\tau)|x_k(\tau), y(\tau)) := \mathcal{N}(x_{k-1}(\tau)|\mu_\theta(x_k(\tau), k, y(\tau)), \sigma_k^2\mathbf{I})$, where $\mathcal{N}$ denotes a Gaussian distribution, and $x_k(\tau) := \{s_0^k, s_1^k, ..., s_T^k\}$ denotes the noisy trajectory at step $k$. We could get such a DD by **training** on a tractable variational evidence lower bound of $\log p_\theta(x_0(\tau)|y(\tau))$ with a surrogate loss (Ho et al., 2020):

$$\mathcal{L}_{\text{denoise}}(\theta) := \mathbb{E}_{k \sim U\{1,...,K\}, x_0 \sim \mathcal{D}, \epsilon \sim \mathcal{N}(\mathbf{0}, \mathbf{I})} \left\| \epsilon - \epsilon_\theta(x_k(\tau), k, y(\tau)) \right\|^2, \tag{4}$$

where $\epsilon$ is the noise added to the data sample $x_0$ to produce $x_k$, and we use a deep neural network $\epsilon_\theta(x_k, k, y(\tau))$ to estimate the noise $\epsilon$.

At a timestep $t$ during **inference**, the diffuser combines the observed states and padding noise as inputs $\tilde{x}_K = \{s_0, ..., s_t, s_{t+1}^K, ..., s_T^K\}$. It then generates a trajectory $\tilde{x}_0 = \{s_0, ..., s_t, \tilde{s}_{t+1}, ..., \tilde{s}_T\}$, which consists of a sequence of multi-step states with expected property $y(\tau)$. Subsequently, an action $a_t$ is typically obtained by inputting the observed states $s_{t-L:t}$ over a horizon $L$ and the generated next state $\tilde{s}_{t+1}$ into an inverse dynamics model $f_\phi$, *i.e.*, $a_t = f_\phi(s_{t-L:t}, \tilde{s}_{t+1})$. An illustration of this process can be seen in Fig. 7. The inverse dynamic model could be independently trained by

$$\mathcal{L}_a(\phi) = \mathbb{E}_{\{s_{t-L:t+1}, a_t\} \sim \mathcal{D}} ||a_t - f_\phi(s_{t-L:t}, s_{t+1})||^2. \tag{5}$$

However, the training objective described in Eq. (4) does not provide explicit guidance on maintaining dynamic legitimacy between adjacent states. This may lead to the generation of subsequent states $\tilde{s}_{>t}$ that are not achievable from the previously observed states $s_{\leq t}$, as demonstrated in Fig. 1 and further supported by the experiments shown in Fig. 2. Such issue can cause the inverse dynamic model to produce suboptimal actions due to these anomalous transitions.

**Related Work.** AdaptDiffuser (AD) (Liang et al., 2023) and MetaDiffuser (Ni et al., 2023) use a forward dynamic model for guiding the generation to pursue consistent trajectories, but poor base model generation can hinder effectiveness and necessitate training revision (Ajay et al., 2023). DiffuserLite (DL) (Dong et al., 2024a) reformulates the training process as generative interpolation by training multiple diffusers to generate segment trajectories, but it is inefficient and could limit overall planning ability. Adopting causal network architectures for training diffusers, such as diffusers based on RNN (Chen et al., 2024) or DiT (Peebles and Xie, 2023) with causal attention (Yang et al., 2021), is closely related to addressing this issue by enforcing the generation of next states based on historical states. **A more detailed discussion to the auto-regressive modeling and masked trajectory modeling can be seen in Appendix M.** Therefore, in this work, we firstly aim to propose an effective solution specifically in the context of auto-bidding for addressing the dynamic legitimacy issue without introducing extra modules.

## 3 CAUSAL AUTO-BIDDING VIA DIFFUSION COMPLETER-ALIGNER

We propose a novel causal auto-bidding method, termed CBD, to enhance the diffuser's bidding performance in large-scale auctions by *i*) reformulating training a diffuser as learning a completer and *ii*) aligning the generated states to advertisers' objectives in the inference process by refining the generation via an aligner. A detailed illustration of our CBD method could be seen in Fig. 1.

## 3.1 COMPLETER: LEARNING TO COMPLETE IN TRAINING

We now provide an insightful analysis of the compounding error that arises from directly applying diffusers for auto-bidding, which leads to the necessity of reformulating the bidding process as a completion process. We denote $\boldsymbol{o}_t := \{\boldsymbol{s}_{\leq t}, \boldsymbol{y}(\tau)\}$ as all the observations and $\boldsymbol{s}'$ as the generated future states. Then, we can represent a simple optimal bidding strategy in inference as $\boldsymbol{a}_t = f_\phi(\boldsymbol{o}_t, \boldsymbol{s}'), \boldsymbol{s}' \sim p_{\theta^*}(\boldsymbol{s}'|\boldsymbol{o}_t, \boldsymbol{z}) = \mu_{\theta^*}(\boldsymbol{z}|\boldsymbol{o}_t) + \sigma_{\theta^*}(\boldsymbol{z}|\boldsymbol{o}_t)\boldsymbol{\epsilon}, \boldsymbol{\epsilon} \sim \mathcal{N}(\mathbf{0}, \mathbf{I})$, where $\boldsymbol{z}$ is the input padding noise that will be denoised to the next states $\boldsymbol{s}'$ conditioned on $\boldsymbol{o}_t$. We have a diffuser trained via Eq. 4, *i.e.*, denoising $\boldsymbol{z}$ to $\boldsymbol{s}'$ conditioned on a diffused observation $\tilde{\boldsymbol{o}}_t = a_t \boldsymbol{o}_t + b_t \boldsymbol{\eta}, \boldsymbol{\eta} \sim \mathcal{N}(\mathbf{0}, \mathbf{I})$. For brevity, we assume $a_t = b_t = 1$. Thus, the trained diffuser is actually performing $p_\theta(\boldsymbol{s}'|\tilde{\boldsymbol{o}}_t, \boldsymbol{z}) = \mu_\theta(\boldsymbol{z}|\boldsymbol{o}_t + \boldsymbol{\eta}) + \sigma_\theta(\boldsymbol{z}|\boldsymbol{o}_t + \boldsymbol{\eta})\boldsymbol{\epsilon} = [\mu_\theta(\boldsymbol{z}|\boldsymbol{o}_t) + d_\mu(\boldsymbol{z}|\boldsymbol{o}_t, \boldsymbol{\eta})] + [\sigma_\theta(\boldsymbol{z}|\boldsymbol{o}_t) + d_\sigma(\boldsymbol{z}|\boldsymbol{o}_t, \boldsymbol{\eta})]\boldsymbol{\epsilon}$, where $d$ denotes the residual part. Now, we can conclude that if the diffuser is well-trained in transitioning to the distribution on $\boldsymbol{s}'$ of optimal value, we have the *distribution matching error*, denoted as $\mathcal{E}_t$, of applying $p_\theta$ under the diffuser inference scheme, *i.e.*,

$$\mathcal{E}_t = \mathbb{D}_{\mathrm{KL}}[p_{\theta^*}(\boldsymbol{s}'|\boldsymbol{o}_t, \boldsymbol{z})\|p_\theta(\boldsymbol{s}'|\boldsymbol{o}_t, \boldsymbol{z})]$$
$$= \frac{1}{2}\left[\frac{\sigma_{\theta^*}(\boldsymbol{z}|\boldsymbol{o}_t)^2 + d_\mu(\boldsymbol{z}|\boldsymbol{o}_t, \boldsymbol{\eta})^2}{(\sigma_{\theta^*}(\boldsymbol{z}|\boldsymbol{o}_t) - d_\sigma(\boldsymbol{z}|\boldsymbol{o}_t, \boldsymbol{\eta}))^2} - 1 + \ln\frac{(\sigma_{\theta^*}(\boldsymbol{z}|\boldsymbol{o}_t) - d_\sigma(\boldsymbol{z}|\boldsymbol{o}_t, \boldsymbol{\eta}))^2}{\sigma_{\theta^*}(\boldsymbol{z}|\boldsymbol{o}_t)^2}\right]. \quad (6)$$

Proof can be seen in Appendix B. Intuitively, this is because of the unnecessary denoising of observations during inference, leading to $\mathcal{E}_t > 0$. As there are multiple steps of a bidding period, the final compounding error could be extremely large. This could lead to suboptimal performance of the diffuser, as it risks deviating from the optimal distribution of next states in $p_{\theta^*}$, potentially entering dynamic illegitimate areas, as illustrated in Fig. 1.

To address this issue in auto-bidding, a practical approach to mitigate the compounding error is to eliminate the discrepancy between training and inference phases. This can be achieved by leveraging observed information to enhance generation, *i.e.*, learning to directly minimize the divergence $\mathbb{D}_{\mathrm{KL}}[p_{\theta^*}(\boldsymbol{s}'|\boldsymbol{o}_t, \boldsymbol{z})\|p_\theta(\boldsymbol{s}'|\boldsymbol{o}_t, \boldsymbol{z})]$. Specifically, the generation procedure of training and inference phase should be unified as follows: at a timestep $t$, we start generation by sampling $\boldsymbol{x}_K(\tau) = \{\boldsymbol{s}_0^K, \boldsymbol{s}_1^K, ..., \boldsymbol{s}_T^K\}$ from $\mathcal{N}(\mathbf{0}, \mathbf{I})$ and assign the observed states $\boldsymbol{s}_{0:t}$ to the corresponding places to get the input of the diffuser, *i.e.*,

$$\tilde{\boldsymbol{x}}_K(\tau, t) = \{\underbrace{\boldsymbol{s}_0, \ldots, \boldsymbol{s}_t}_{\text{Query}}, \underbrace{\boldsymbol{s}_{t+1}^K, \ldots, \boldsymbol{s}_T^K}_{\text{padding with noise}}\}. \quad (7)$$

Then, according to denoising process $p_\theta(\boldsymbol{x}_{k-1}(\tau)|\boldsymbol{x}_k(\tau), \boldsymbol{y}(\tau))$, we could sample the predicted next step's trajectory by

$$\tilde{\boldsymbol{x}}_{k-1}(\tau, t) = \mu_\theta(\tilde{\boldsymbol{x}}_k(\tau, t), k, \boldsymbol{y}(\tau)) + \sigma_k \boldsymbol{z}, \quad (8)$$

where $\boldsymbol{z} \sim \mathcal{N}(\mathbf{0}, \mathbf{I})$. The $\sigma_k$ could be typically set as $\sigma_k := 1 - \alpha_k$ and the $\mu_\theta$ could be parameterized by $\mu_\theta(\tilde{\boldsymbol{x}}_k(\tau, t), k, \boldsymbol{y}(\tau)) = \frac{1}{\sqrt{\alpha_k}}(\tilde{\boldsymbol{x}}_k(\tau, t) - \frac{1-\alpha_k}{\sqrt{1-\bar{\alpha}_k}}\hat{\boldsymbol{\epsilon}}_k)$, where $\bar{\alpha}_k := \prod_{i=1}^k \alpha_k$ and the $\hat{\boldsymbol{\epsilon}}_k$ could be modeled with a conditional model $\epsilon_\theta(\tilde{\boldsymbol{x}}_k(\tau, t), k, \boldsymbol{y}(\tau))$ by randomly dropping the condition $\boldsymbol{y}(\tau)$ in a classifier-free guidance scheme (Ho and Salimans, 2022). By recursively applying the above procedure and assigning the first $t$-length of the generation with the query, we could get the final generated trajectory

$$\tilde{\boldsymbol{x}}_0(\tau, t) = \{\underbrace{\boldsymbol{s}_0, \ldots, \boldsymbol{s}_t}_{\text{Query}}, \underbrace{\tilde{\boldsymbol{s}}_{t+1}, \ldots, \tilde{\boldsymbol{s}}_T}_{\text{Answer}}\}. \quad (9)$$

Additionally, as the dynamic legitimacy issue may occur in every position of the generated trajectory, we augment the previous diffuser training procedure by setting $t$ as an extra random variable to construct the observations query and reformulate training as learning to complete. Specifically, with the augmented random variable $t$, the completer $p_\theta$ should capture the causality between the observed part $\{\boldsymbol{s}_0, ..., \boldsymbol{s}_t\}$ and the remaining part $\{\boldsymbol{s}_{t+1}, ..., \boldsymbol{s}_T\}$ with an "instruction" $\boldsymbol{y}(\tau)$, resembling a query-and-answer style like LLM. Therefore, the training objective is changed into

$$\max_\theta \mathbb{E}_{\tau \sim \mathcal{D}}[\log p_\theta(\boldsymbol{s}_{t+1:T}|\boldsymbol{s}_{0:t}, \boldsymbol{y}(\tau))]. \quad (10)$$

To achieve this objective, we can make a small modification by introducing an augmented new completion training loss $\mathcal{L}_c$, while still utilizing the powerful generation ability of diffusers, *i.e.*,

$$\mathcal{L}_c(\theta) := \mathbb{E}_{k \sim U\{1, ..., K\}, \boldsymbol{x}_0 \sim \mathcal{D}, t \sim U\{0, ..., T-1\}, \boldsymbol{\epsilon} \sim \mathcal{N}(\mathbf{0}, \mathbf{I})}\|\boldsymbol{\epsilon} - \epsilon_\theta(\tilde{\boldsymbol{x}}_k(\tau, t), k, \boldsymbol{y}(\tau))\|_{t+1:T}^2 \quad (11)$$

where $\|\cdot\|_{t+1:T}^2$ means we only compute the loss on the position index $t+1$ to $T$ in a trajectory. With such an additional variable $t \sim \text{Uniform}\{0, ..., T-1\}$ and replacing $\boldsymbol{x}_k(\tau)$ to $\tilde{\boldsymbol{x}}_k(\tau, t)$ of Eq. (7), the model is learned to be aware of the dynamic legitimacy of adjacent states in every positions of the trajectory and thus enhancing the dynamic legitimacy of the whole trajectory.

## 3.2 ALIGNER: REFINING GENERATIONS IN INFERENCE

After training as a completer, the learned bidding policy, which includes the diffuser $p_\theta$ and the inverse dynamic model $f_\phi$, is capable of producing actions with improved dynamic legitimacy. However, its alignment with the expected advertiser preferences, which could be represented by $y(\tau)$, may still be suboptimal. This misalignment can be attributed to factors such as the limited capacity of the model, as we cannot get a perfect model, and the dynamic nature of advertiser preferences, which may include previously unseen preferences in datasets. Additionally, it remains challenging to explicitly control the sampling process during denoising to precisely obtain the expected optimal action $a_t^*$ (Zheng et al., 2023a;b). Consequently, these concerns may lead to misalignment issues, failing to meet the advertiser's expected properties, as illustrated in Fig. 2.

To enhance the quality of generated trajectories and improve subsequent auto-bidding performance, we can leverage a trajectory-level return model to introduce an aligner during the inference stage for refining the generated trajectories. The return model could be independently trained to regress on the conditions $\boldsymbol{y}(\tau)$ in the datasets expressed as

$$\mathcal{L}_r(\varphi) := \mathbb{E}_{\{\boldsymbol{x}_0, \boldsymbol{y}\} \sim \mathcal{D}} || R_\varphi(\boldsymbol{x}_0(\tau)) - \boldsymbol{y}(\tau) ||^2. \tag{12}$$

This regression objective is not restricted by the issue of sparse rewards. Although the immediate reward $r_t$ for most steps may be zero, leading to a sparse reward challenge, the accumulated rewards $\sum_{t=0:T} r_t$ are not zero and can be easy to learn.

There are several options for implementing the aligner scheme. One approach is a best-of-N sampling method that leverages the diverse generation ability of diffusers. This method generates multiple future trajectories in parallel and employs the trajectory-level return model to select the best trajectory, as illustrated below:

$$\tilde{\boldsymbol{s}}'_{t+1:T} \leftarrow \underset{\tilde{\boldsymbol{s}}^i_{t+1:T}}{\arg\min} || R_\varphi(\boldsymbol{s}_{0:t}, \tilde{\boldsymbol{s}}^i_{t+1:T}) - \boldsymbol{y}(\tau) ||^2, i = 1 \sim N. \tag{13}$$

Alternatively, the aligner can be implemented using a more efficient revision method that performs a gradient update on $\tilde{\boldsymbol{s}}_{t+1:T}$ after the generation, which could be expressed as

$$\tilde{\boldsymbol{s}}'_{t+1:T} \leftarrow \tilde{\boldsymbol{s}}_{t+1:T} - \lambda \nabla_{\tilde{\boldsymbol{s}}_{t+1:T}} || R_\varphi(\tilde{\boldsymbol{x}}_0(\tau, t)) - \boldsymbol{y}(\tau) ||^2. \tag{14}$$

When the generated trajectory has a higher estimated property value $R_\varphi(\tilde{\boldsymbol{x}}_0(\tau, t))$ than the expected property $\boldsymbol{y}(\tau)$ provided by the advertiser, the refinement operation would adjust the trajectory to a new one with a lower property value, and vice versa. Note that this approach differs from the classifier-guidance denoising technique used in diffusers, which iteratively interacts with every denoising step by $\boldsymbol{x}_{k-1} \sim \mathcal{N}(\boldsymbol{\mu}_k + \alpha \nabla_\theta \mathcal{J}(\boldsymbol{\mu}_k), \sigma_k^2 \boldsymbol{I})$. This iterative interaction creates challenges in extending it to various denoising sampling methods, such as DDIM (Song et al., 2021) and ODE (Zheng et al., 2023a) methods. In contrast, our approach refines only the final output to maintain computational efficiency and enable flexible future development, regardless of the training and inference techniques employed by the diffuser. More implementations of the aligner scheme could include multi-round refinement or exploring non-gradient methods, such as directly adding a residual to the output like LoRA (Hu et al., 2022) for optimal alignment. However, in this work, we found the gradient-based method in Eq. 14 to be effective and robust enough. We will report the aligner's performance based on this gradient-based refinement method.

Finally, we feed the updated $\tilde{\boldsymbol{s}}'_{t+1}$ into the inverse dynamic model to obtain the final action. For a better understanding of our CBD method, we provide the detailed training and inference procedure in Algorithm 1 and 2 in Appendix D.

## 4 EXPERIMENT

### 4.1 SETUP

**Datasets.** To evaluate the performance of bidding strategy decision-making in large-scale ad auctions, we employ AuctionNet and its sparse variant, a publicly available benchmark from Alibaba designed for ad auctions, based on a real-world online advertising platform (Su et al., 2024). Each dataset comprises 21 advertising delivery periods, with each period containing approximately 500,000 impression opportunities, divided into 48 intervals. Details of the AuctionNet are in Appendix C.

Table 1: Comparison with baselines under the metric of Value in different budget settings in the MCB task. The best results are bolded and the base policy model's results are underlined to demonstrate the performance improvement of our CBD method in the last column (improvements are statistically significant, *i.e.*, two-sided t-test with $p < 0.05$).

| Dataset | Budget | RL | | | | Transformer | | | Diffuser | | | | | *improve* |
|---|---|---|---|---|---|---|---|---|---|---|---|---|---|---|
| | | USCB | BCQ | CQL | IQL | DT | CDT | DT-S | DiffBid | DiT | DiT-causal | CBD-Completer | CBD | |
| AuctionNet | 50% | 133.1 | 184.2 | 180.1 | 185.0 | 186.2 | 190.3 | 196.8 | 161.9 | 156.4 | 160.0 | 191.9 | **198.6**±1.8 | +22.3% |
| | 75% | 201.9 | 260.1 | 256.6 | 259.9 | 234.1 | 290.8 | 298.0 | 233.7 | 228.3 | 229.1 | 280.3 | **298.3**±2.6 | +27.8% |
| | 100% | 267.4 | 270.6 | 336.6 | 322.2 | 364.0 | 366.5 | 373.1 | 316.5 | 307.0 | 311.8 | 370.4 | **374.0**±3.0 | +18.3% |
| | 125% | 335.0 | 333.0 | 400.2 | 398.6 | 393.0 | 400.2 | 418.6 | 384.0 | 375.1 | 381.8 | 420.5 | **427.2**±4.0 | +11.1% |
| | 150% | 415.9 | 388.5 | 465.3 | 457.3 | 445.2 | 429.5 | 437.6 | 452.5 | 435.1 | 440.6 | 473.7 | **480.5**±4.6 | +6.19% |
| AuctionNet-sparse | 50% | 15.42 | 18.10 | 16.79 | 20.21 | 18.20 | 18.01 | 19.62 | 14.54 | 14.12 | 14.33 | 20.08 | **20.56**±0.21 | +41.4% |
| | 75% | 21.63 | 27.00 | 21.80 | 26.90 | 27.50 | 27.54 | 28.70 | 23.25 | 22.58 | 22.60 | 29.90 | **29.97**±0.26 | +28.9% |
| | 100% | 28.48 | 30.54 | 30.15 | 36.06 | 31.30 | 34.31 | 36.82 | 31.33 | 31.02 | 31.27 | 40.40 | **40.71**±0.31 | +29.9% |
| | 125% | 34.88 | 35.19 | 37.67 | 43.92 | 43.51 | 42.52 | 43.91 | 40.54 | 39.40 | 39.14 | 45.54 | **46.04**±0.45 | +13.5% |
| | 150% | 43.08 | 36.97 | 44.43 | 49.27 | 50.04 | 50.93 | 49.65 | 47.18 | 45.81 | 46.07 | 51.10 | **51.35**±0.47 | +8.91% |

**Evaluation Metrics.** We test the baselines and our method on the maximize conversions bidding (MCB) task[1], and we adopt three metrics to evaluate the performance:

- **Value**: the total received impression value, *i.e.*, number of conventions $\sum_i o_i v_i$;

- **ER** (Exceeding rate of the KPI constraints): In a specific cost-cap setting, advertisers are concerned with how closely the strategy's actual KPI performance matches the target KPI limit, such as CPA. To assess this, we introduce the ER metric for a delivery period, defined by $ER = \frac{1}{J}\sum_j C_j^{real}/C_j = \frac{1}{J}\sum_j(\sum_i c_{ij}o_i/\sum_i p_{ij}o_i)/C_j$.

- **Score**: by introducing a penalty term $penaty_j = min\{(\frac{C_j}{\Sigma_i c_{ij}o_i/\Sigma_i p_{ij}o_i})^2, 1\}$, we can get a score as $score = (\sum_i o_i v_i) \times min\{penaty_j\}_{j=1\sim J}$.

**Baselines.** We conduct a comprehensive comparison of our method against a variety of baselines, including offline RL, DTs, and diffusers, to assess the potential of CBD as a new generative backbone. For RL, we evaluate against **USCB** (He et al., 2021), **BCQ** (Fujimoto et al., 2019), **CQL** (Kumar et al., 2020), and **IQL** (Kostrikov et al., 2022). For DTs, our comparisons include **DT** (Chen et al., 2021), **CDT** (Liu et al., 2023), and **DT**-S, which is a DT-based approach that incorporates the reward function by considering both the winning value and KPI constraints. **DT**-S represents the most advanced DT backbone utilized in prior works such as GAS (Li et al., 2024) and GAVE (Gao et al., 2025a). For diffusers, we compare with **DiffBid** (Guo et al., 2024), **DiT** (Peebles and Xie, 2023), which enhances DiffBid by replacing its U-Net (Ronneberger et al., 2015) backbone with a diffusion transformer, and **DiT-causal**, which further refines DiT by integrating a causal-attention mechanism (Waswani et al., 2017; Yang et al., 2021). The diffuser backbone of CBD is also based on Diffbid. For each method, the reported results are averaged over 5 random seeds.

We move the **implementation details** and detailed experimental setup to Appendix C.

## 4.2 Performance Comparison with Baselines

In this experiment, we conducted a performance comparison of various baseline methods under different settings, including varying datasets and budget constraints in the MCB task. The results are indicated by the value as a comprehensive assessment of the performance, as presented in Table 1. The CBD method consistently outperforms other models across all budget levels, with the most notable improvements in lower budget and sparse reward scenarios. Even without employing a trajectory-level return model for updating the predicted next state (referred to as CBD-Completer), its performance still significantly exceeds the baseline Diffbid, highlighting the benefits of employing a completer to address the dynamic legitimacy issue. Additionally, the DiT and its variant DiT-causal do not enhance the diffuser's performance, underscoring the effectiveness of our training objective, *i.e.*, learning to perform auto-bidding as learning to complete. This insight could be inspiring for future research. To better understand the generation uncertainty issue, we provide a visualization in Fig. 2 and in Appendix in Appendix K by randomly sampling trajectories from the test set and comparing the generated trajectories between DiffBid and our methods. To enable a more comprehensive comparison with diffusion-based methods across a wider range of metrics, specifically considering the KPI constraints common in the cost-cap bidding mode, we present the

---

[1]https://support.google.com/google-ads/answer/7381968?hl=en

Table 2: Comparison under more metrics of the diffuser-based methods and CBD variants, where arrow directions indicate performance improvement direction.

| Dataset | Metrics | Diffbid | DiT | DiT-causal | CBD-Transformer | CBD-CVAE | CBD-$\mathcal{A}$ | CBD-Distillation | CBD-Completer | CBD |
|---|---|---|---|---|---|---|---|---|---|---|
| AuctionNet | Value ↑ | 316 | 307 | 311 | 313 | 314 | 354 | 365 | 370 | **374** |
| | ER ↓ | 1.37 | 1.42 | 1.40 | 1.35 | **0.90** | 1.13 | 1.13 | 1.12 | 1.10 |
| | Score ↑ | 214 | 196 | 205 | 210 | 284 | 294 | 285 | 292 | **298** |
| AuctionNet-sparse | Value ↑ | 31.3 | 31.0 | 31.2 | 31.2 | 31.2 | 32.0 | 39.8 | 40.4 | **40.7** |
| | ER ↓ | 1.23 | 1.32 | 1.30 | 1.21 | 1.25 | 1.24 | 0.95 | 0.90 | **0.86** |
| | Score ↑ | 23.1 | 22.0 | 22.5 | 21.9 | 29.2 | 29.5 | 34.0 | 35.5 | **37.0** |

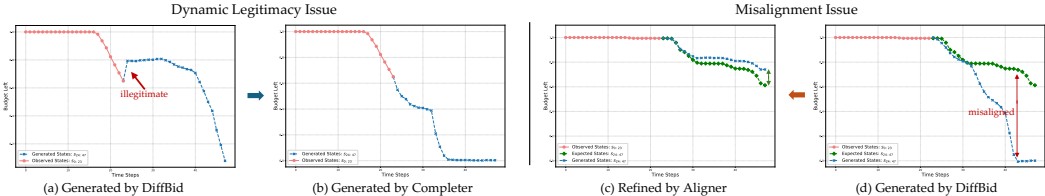

(a) Generated by DiffBid    (b) Generated by Completer    (c) Refined by Aligner    (d) Generated by DiffBid

Figure 2: visualization of the generation uncertainty issue using an example case. In all subfigures, the red and green lines depict the ground-truth state information of the remaining budget, and the blue lines represent the generation results. Observing from $t = 0$ to 23, DiffBid generates states where the remaining budget at $t = 24$ is greater than at $t = 23$, indicating a **dynamic legitimacy issue**. Additionally, DiffBid could have significant deviations of the generated trajectories from the expected trajectory, highlighting a **misalignment issue**.

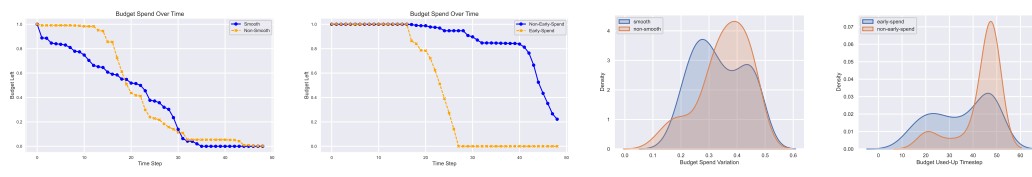

(a) Align on Smoothness    (b) Align on Early-Spend    (c) Smoothness Dist.    (d) Early-Spend Dist.

Figure 3: Results for property alignment. (a-b) show the generated trajectories with different properties, including smoothness and early spending; (c-d) illustrate the property distribution after aligning the generated trajectories to the desired properties at each bidding timestep.

results in Table 2. The CBD method consistently exhibits superior performance in ER, suggesting a more cost-efficient bidding strategy. Overall, CBD excels particularly in sparse scenarios, strongly endorsing the diffuser-based CBD as an exceptional generative backbone for industrial auto-bidding, where conversions are typically sparse.

### 4.3 PROPERTY ALIGNMENT PERFORMANCE

In addition to the aligner's improvements shown in Table 1, we also examine other preferences depicted in Figures 3(a) and 3(b). We can generate diverse trajectories and utilize the aligner to refine these trajectories with desired characteristics, such as achieving a smoother trajectory alignment. Specifically, Figure 3(c) illustrates the distribution of budget spending variation, which represents smoothness; the lower the variation, the smoother the strategy. Figure 3(d) displays the distribution of budget used-up timesteps to demonstrate the property of early spending. The smooth budget spending is more concentrated and exhibits low variation, while the early-spend (depicted in blue) and non-early-spend (depicted in orange) differ in their timestep distributions. These results highlight the aligner's effectiveness in refining trajectories. This flexibility is a significant advantage of the planning-based auto-bidding strategy, as it enhances the reliability of the bidding process and aids in diagnosing and correcting errors or biases.

### 4.4 ABLATION STUDY

**Alternative Backbones compared to Diffusion Model.** We conducted a comparison experiment with other simpler generative models, including a Transformer-based CBD method and a Conditional VAE (CVAE)-based CBD method, noted as CBD-Transformer and CBD-CVAE, respectively. Specifically, the CBD-Transformer utilizes a transformer where a masked sequence of states serves as the input, and the return of the trajectory acts as a condition interacting with the transformer block via AdaLN-zero. The CBD-CVAE employs a Temporal-UNet as the encoder and decoder, using the returns as the

condition. The results in Table 2 clearly demonstrate that employing diffusion models can achieve a better performance, demonstrating the necessity of adopting diffusion models for CBD.

**Diffusion Steps and Distillation.** The diffusion model typically requires repeating the recursive denoising step in Eq. (8) over multiple iterations, often up to 1000 steps. Therefore, we examined how the number of denoising steps affects performance to determine if a larger number of steps could improve outcomes. As shown in Fig. 4, we trained the diffusion model with varying denoising steps, including 10, 50, 100, 500, and 1000.

Our findings indicate that 100 denoising steps are sufficient for achieving good performance, resulting in a computationally efficient approach. Though industrial auto-bidding services have sufficient time for model's response as stated in Section 2.1, we also introduce a distillation method as a post-training trick, named SiD (Zhou et al., 2024), to distill our trained diffusion model into a one-step generator using the same U-Net while maintaining similar performance as the results of CBD-Distillation shown in Table 2. As these acceleration techniques can be directly applied to our method, the inference latency can be significantly reduced. Therefore, we highlight the importance of identifying the root causes of failures and developing advanced methods for training the original diffusion models for auto-bidding, which is the urgent need in this field.

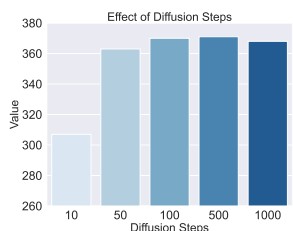

Figure 4: Effect of the diffusion steps.

We provide more ablation studies regarding the **Aligner's Robustness** in Appendix E, **Necessity of Inverse Dynamic Model** in Appendix F, and **Extension to Offline RL Tasks** in Appendix G.

## 5 ONLINE A/B TEST

To verify the effectiveness of CBD, we have deployed it to an online ad platform. Under the MCB setting, the advertisers set the budget with or without the CPA/ROI constraint, and the bidding strategy aims to get as many conversions as possible under constraints. We compared CBD with the in-production DT-based method of a dedicated reward design. Over 7 days A/B test, there have been an average of 14700 campaigns per day. With the same budget allocated for each method's experiment, each method achieved a daily average of approximately 80,000,000 impressions, which is substantial enough to demonstrate the effectiveness of the methods.

As shown in Table 3, the average achieved conversions (Target Cost) for advertisers are improved by an average of +2.0% while spending a similar budget (Cost), and the CPA valid ratio is also improved by +1.85%, which are significant improvements for such a large-scale auto-bidding production. Though our method incurs an additional 6ms compared to DT (6ms), it is still acceptable and cost-effective without needing additional computational resources, which could support a maximum of 26ms at the same frequency (around 20s) of calling the service to update the bidding parameters in the current computational resource. Given the substantial commercial value achieved, the additional inference latency is justifiable.

Table 3: A/B test results.

| Metric | -*Compare* |
|---|---|
| Inference Latency | +6 ms |
| Cost | +0.0% |
| Target Cost | +2.0% |
| CPA Valid Ratio | +1.85% |

## 6 LIMITATIONS AND CONCLUSION

This paper focuses exclusively on the auto-bidding strategy, which represents only one aspect of the real-time bidding system. Therefore, it is a limitation that we do not deeply consider other components, such as auction mechanism design and cooperation between advertisers, and their influence on the auto-bidding strategy. Given the difficulties in replicating these components within a public benchmark, we believe the effectiveness of CBD in handling real-world complexity could be empirically supported by the online results.

In conclusion, this paper explores the potential of diffusers for industrial auto-bidding. After analyzing failures in applying diffusers, we identified generation uncertainty as a key root cause and introduced a causal auto-bidding method based on a novel diffusion completer-aligner framework. Our method addresses the dynamic legitimacy issue through the completer and refines the generations to address misalignment using an aligner. Experiments support our analysis and demonstrate the superiority of CBD as a generative backbone for auto-bidding.

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

APPENDIX

# A  DETAILED BACKGROUND

## A.1  AUTO-BIDDING IN COMPUTATIONAL ADVERTISING

The rapid digital transformation of commerce has significantly expanded the reach of online advertising platforms, making them essential for engaging audiences and driving sales (Wang and Yuan, 2015; Evans, 2009). As reported by the International Advertising Bureau, U.S. digital ad revenue from computational advertising revenue has reached \$114.2 billion (Bureau, 2024), significantly contributing to the success and sustainability of information systems and technologies. The primary goal of computational advertising is to effectively match ads with their relevant contexts on the web, such as the content where the ad is displayed and users' search queries or browsing behavior (Dave and Varma, 2014; Wang et al., 2017). With the continuous influx of ad impression opportunities to the platform, real-time bidding (RTB) based display advertising, which emerged in 2009 (Muthukrishnan, 2009), has become a major paradigm of computational advertising, which enables online advertising platforms to sell individual ad impressions via hosting a real-time auction and facilitates advertisers to bid for the impression opportunity based on estimated value. Specifically, in real-time auto-bidding (Ren et al., 2017), when an impression opportunity is generated from a user's visits, a bid request for the ad display impression opportunity, including its features like user, context, and auction status, is sent to numerous advertisers via an ad exchange. Then, advertisers employ specific auto-bidding algorithms to assess the bid request's potential value and determine a bid price via Eq. 2 in real-time on demand-side platforms (DSPs). **Please note that the bidding parameters in Eq. 2 are adjusted by auto-bidding methods at a low frequency, such as every 30 minutes, rather than for every single impression that arises in milliseconds.** The ad exchanger then selects the highest bidder to display the ad, charging them the market prices, typically the second-highest bid price in a second-price auction setting (Amin et al., 2012). A detailed illustration can be seen in Figure 5.

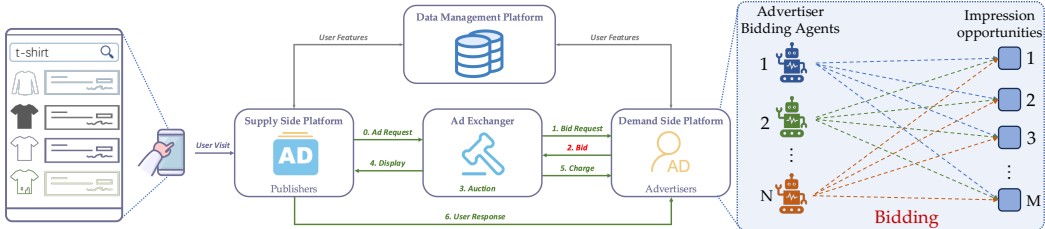

Figure 5: Real-time bidding system.

To maximize the total value of impressions for an advertiser while adhering to certain economic constraints, research into advanced auto-bidding strategies has progressed from rule-based approaches to prediction-based methods, and more recently, to reinforcement learning-based techniques (Ou et al., 2024). For rule-based methods, the proportional-integral-differential (PID) is the most widely used controller method (Bennett, 1993). The control signal consists of proportional, integral, and differential terms of the error, accounting for current, past, and future trends, resulting in comprehensive control of the variable. A typical example of applying PID in online advertising is controlling the campaign's Cost-Per-Action (CPA) to a reference level (Chen et al., 2011). The controller could also be used to control the budget pacing to spend the budget smoothly (Grislain et al., 2019; Myerson, 1981). In addition to the PID controller, custom designs exist to adjust bids based on feedback data by classifying states as comfortable or risky and updating controlled variables accordingly (Geyik et al., 2016), while Multi-KPI (Kitts et al., 2017) controlled multiple KPIs simultaneously based on observed and target errors. For prediction-based techniques integrating auto-bidding strategies with upstream prediction tasks, such as CTR prediction, CVR prediction, and market price prediction, bid optimization is no longer an independent task but is correlated with these prediction tasks and jointly optimized. It is because a previous work, Bidding Machine (Ren et al., 2017), argues that interconnected tasks and sequential optimization lead to suboptimal solutions, and joint optimization allows the learning process to focus on valuable and competitive impressions. Some methods have tried this view, such as an EM-like (Expectation-Maximization) algorithm proposed to iteratively

optimize the CTR learning and bid optimization (Ren et al., 2016) and its extension incorporates market price estimation into the framework (Ren et al., 2017). Besides joint optimization, some methods tried introducing the lift/attribution prediction into the bid optimization, which is to predict the difference value before and after the ad is displayed (li et al., 2018; Ren et al., 2018; Moriwaki et al., 2020; Xu et al., 2016). For the reinforcement learning based methods, they repeatedly interact with the environment, allowing bidding agents to gain value from winning ad impression opportunities and update their subsequent strategies. The entire bidding process for advertisers is essentially a sequential decision-making process, which has recently demonstrated superior effectiveness.

### A.2 SEQUENTIAL DECISION-MAKING FOR AUTO-BIDDING

As the complexity of online bidding environments increased, reinforcement learning (RL) algorithms like USCB (He et al., 2021), SORL (Mou et al., 2022), and MAAB (Wen et al., 2022) became essential for bidding. USCB (He et al., 2021) maximizes the value under multiple constraints and discovers a recursive structure to accelerate convergence. MAAB (Wen et al., 2022) considers the multi-agent modeling to maximize value and social welfare, proposing to mix cooperation and competition among multiple advertisers and improving the platform's revenue. SORL (Mou et al., 2022) proposes a sustainable online RL framework that trains the auto-bidding policy by directly interacting with the RTB system, using safe and efficient online exploration instead of a simulated environment. However, online RL poses risks to the RTB system and cannot fully leverage the extensive historical bidding logs, resulting in a preference for bidding methods based on offline RL. Offline RL methods, which derive policies from existing datasets without requiring online interaction, have demonstrated considerable success. Noteworthy approaches include BCQ (Fujimoto et al., 2019), which constrains the action space to encourage on-policy behavior; CQL (Kumar et al., 2020), which regularizes Q-values for conservative estimates; and IQL (Kostrikov et al., 2022), which facilitates multi-step dynamic programming updates without querying Q-values of out-of-sample actions during training.

Despite their effectiveness, these methods are limited by the Markov Decision Process (MDP) assumption, whereas generative models (Doersch, 2016; Goodfellow et al., 2020; Pan et al., 2023; Ho et al., 2020; Chen et al., 2021) offer greater potential for bidding. Some methods try to apply decision transformers (DTs) (Chen et al., 2021; Liu et al., 2024; Gao et al., 2025b) for auto-bidding. GAS (Li et al., 2024) and GAVE (Gao et al., 2025a) propose data augmentation techniques via introducing an extra value prediction module to enhance the quality of the offline datasets. However, they still rely on dense sparse reward signals, and the introduced value function could suffer from instability issues, making them limited in the industrial auto-bidding tasks.(Ajay et al., 2023). As this paper focuses on proposing a superior generative backbone, we compare our CBD method to DT and its variants directly. In contrast, DiffBid (Guo et al., 2024) employs a decision diffuser to generate a long trajectory based on conditions like return, and then uses an inverse dynamic model to output the final action. However, in large-scale auctions, DiffBid faces challenges related to generation uncertainty, which is the primary focus of our work.

## B PROOF OF THE DISTRIBUTION MISMATCH ERROR

To facilitate understanding the motivation behind the diffusion completer, we provide a detailed derivation of the distribution matching error in this section.

We denote $o_t := \{s_{\leq t}, y(\tau)\}$ as all the observations and $s'$ as the generated future states. As there are multiple denoising steps of a diffusion model, for simplicity to get an insightful analysis, we assume the number of the denoising step as 1. Then, we can represent a simple optimal bidding strategy in inference as

$$a_t = f_\phi(o_t, s'), s' \sim p_{\theta^*}(s'|o_t, z) = \mu_{\theta^*}(z|o_t) + \sigma_{\theta^*}(z|o_t)\epsilon, \epsilon \sim \mathcal{N}(0, I), \quad (15)$$

where $z$ is the input padding noise that will be denoised to the next states $s'$ conditioned on $o_t$. The optimal distribution $p_{\theta^*}(s'|o_t, z)$ signifies that the generated $s'$ will lead to optimal auto-bidding performance.

We have a diffuser trained via Eq. 4, *i.e.*, denoising $z$ to $s'$ conditioned on a diffused observation $\tilde{o}_t = a_t o_t + b_t \eta, \eta \sim \mathcal{N}(0, I)$. For brevity, we assume $a_t = b_t = 1$. Thus, the trained diffuser is

actually performing

$$p_\theta(s'|\tilde{o}_t, z) = \mu_\theta(z|o_t + \eta) + \sigma_\theta(z|o_t + \eta)\epsilon$$
$$= [\mu_\theta(z|o_t) + d_\mu(z|o_t, \eta)] + [\sigma_\theta(z|o_t) + d_\sigma(z|o_t, \eta)]\epsilon, \quad (16)$$

where $d$ denotes the residual part. Intuitively, $p_\theta$ learns to denoise the diffused $\tilde{o}_t$ into clean $o_t$ first, then project $o_t$ to $s'$. This approach could be effective in image generation tasks, where we typically start generation from a completely uninformative prior distribution, such as a Gaussian distribution, because we cannot observe any part of the generated images before generation. However, the first step may be unnecessary in the context of auto-bidding, as we can obtain a portion of the trajectory, *i.e.*, $s_{0:t}$, during inference. Therefore, denoising $s_{0:t}$ could be redundant and might even hinder the utilization of historical information.

Now, we can conclude that if the diffuser is well-trained in transitioning to the distribution on $s'$ of optimal value, *i.e.*,

$$\mathbb{D}_{\mathrm{KL}}[p_{\theta^*}(s'|o_t, z)\|p_\theta(s'|\tilde{o}_t, z)] = 0, \quad (17)$$

we can have a solution that

$$\mu_{\theta^*}(z|o_t) = \mu_\theta(z|\tilde{o}_t) = \mu_\theta(z|o_t) + d_\mu(z|o_t, \eta) \quad (18)$$
$$\sigma_{\theta^*}(z|o_t) = \sigma_\theta(z|\tilde{o}_t) = \sigma_\theta(z|o_t) + d_\sigma(z|o_t, \eta). \quad (19)$$

With this solution, we apply the trained diffuser to the inference stage by receiving the observation $o_t$ as inputs, *i.e.*,

$$p_\theta(s'|o_t, z) = \mu_\theta(z|o_t) + \sigma_\theta(z|o_t)\epsilon$$
$$= [\mu_{\theta^*}(z|o_t) - d_\mu(z|o_t, \eta)] + [\sigma_{\theta^*}(z|o_t) - d_\sigma(z|o_t, \eta)]\epsilon. \quad (20)$$

Finally, we can get the *distribution matching error*, denoted as $\mathcal{E}_t$, of applying $p_\theta$ under the diffuser inference scheme, *i.e.*,

$$\mathcal{E}_t = \mathbb{D}_{\mathrm{KL}}[p_{\theta^*}(s'|o_t, z)\|p_\theta(s'|o_t, z)]$$
$$= \frac{1}{2}\left[\frac{\sigma_{\theta^*}(z|o_t)^2 + d_\mu(z|o_t, \eta)^2}{(\sigma_{\theta^*}(z|o_t) - d_\sigma(z|o_t, \eta))^2} - 1 + \ln\frac{(\sigma_{\theta^*}(z|o_t) - d_\sigma(z|o_t, \eta))^2}{\sigma_{\theta^*}(z|o_t)^2}\right] > 0. \quad (21)$$

The $\mathcal{E}_t$ can only be 0 when $d_\mu = d_\sigma = 0, \forall \eta \sim \mathcal{N}(0, I)$, which intuitively means the trained diffuser is ideal in denoising any $\tilde{o}_t$ with any noise $\eta$ to $o_t$. However, this is difficult and even impossible with the existing training methods for diffusers. Actually, this could be unnecessary, as we can model the training as a completion task to directly minimize the divergence $\mathbb{D}_{\mathrm{KL}}[p_{\theta^*}(s'|o_t, z)\|p_\theta(s'|o_t, z)]$.

# C   DETAILS OF EXPERIMENTS

**Datasets.** AuctionNet effectively simulates the complexity and integrity of real-world ad auctions through the interaction of several key modules: the ad opportunity generation module, the auto-bidding module, and the auction mechanism module. The AuctionNet benchmark comprises two datasets: AuctionNet and AuctionNet-sparse. AuctionNet-sparse is a sparser version of AuctionNet, featuring fewer conversions. Each dataset includes 21 advertising delivery periods, with each period containing approximately 500,000 impression opportunities, divided into 48 intervals. The models are trained on these two datasets, while 5,000 randomly selected trajectories from each dataset are used exclusively for evaluating the visualization of the generation. Detailed parameters are provided in Table 4. Additionally, the state at each time step includes the following information:

- time_left: The remaining time steps left in the current advertising period.
- budget_left: The remaining budget that the advertiser has available to spend in the current advertising period.
- historical_bid_mean: The average values of bids made by the advertiser over past time steps.
- last_three_bid_mean: The average values of bids over the last three time steps.
- historical_LeastWinningCost_mean: The average of the least cost required to win an impression over previous time steps.

- historical_pValues_mean: The average of historical p-values over past time steps.
- historical_conversion_mean: The average number of conversions (e.g., sales, clicks, etc.) the advertiser achieved in previous time steps.
- historical_xi_mean: The average winning status of advertisers in impression opportunities, where 1 represents winning and 0 represents not winning.
- last_three_LeastWinningCost_mean: The average of the least winning costs over the last three time steps.
- last_three_pValues_mean: The average of conversion probability of advertising exposure to users over the last three time steps.
- last_three_conversion_mean: The average number of conversions over the last three time steps.
- last_three_xi_mean: The average winning status of advertisers over the last three time steps.
- current_pValues_mean: The mean of p-values at the current time step.
- current_pv_num: The number of impressions served at the current time step
- last_three_pv_num_total: The total number of impressions served over the last three time steps.
- historical_pv_num_total: The total number of impressions served over past time steps.

Table 4: The parameters of AuctionNet and AuctionNet-sparse.

| Params | AuctionNet | AuctionNet-Sparse |
|---|---|---|
| Trajectories | 479,376 | 479,376 |
| Delivery Periods | 9,987 | 9,987 |
| Time steps in a trajectory | 48 | 48 |
| State dimension | 16 | 16 |
| Action dimension | 1 | 1 |
| Action range | [0, 493] | [0, 8178] |
| Impression's value range | [0, 1] | [0, 1] |
| CPA range | [6, 12] | [60, 130] |
| Total conversion range | [0, 1512] | [0, 57] |

**Baseline Details.** For RL methods, **USCB** (He et al., 2021), an online RL approach that adjusts parameters dynamically to the optimal bids; **BCQ** (Fujimoto et al., 2019), a typical offline RL method that updates policies with only access to a fixed dataset; **CQL** (Kumar et al., 2020), an offline RL method for learning conservative value function by regularizing Q-values; **IQL** (Kostrikov et al., 2022), an offline RL method that enables multi-step dynamic programming updates without querying out-of-sample actions;

**Evaluation Details.** To assess the auto-bidding performance, we perform experiments in a simulated environment that mirrors a real-world advertising system, as provided by Alibaba (Su et al., 2024). During the evaluation, an episode—also known as an advertising delivery period—consists of a day divided into 48 intervals, each lasting 30 minutes. Each episode includes approximately 500,000 impression opportunities that occur sequentially. An advertising delivery period features 48 advertisers from various categories, each with distinct budgets and CPAs, competing for all impression opportunities during the period. In each evaluation, our well-trained model acts as a specific advertiser, participating in bidding with a given budget and CPA. To thoroughly evaluate the model's performance across different advertisers, we employ various advertiser configurations and advertising periods, conducting multiple evaluations in the simulated environment and averaging the results to obtain the evaluation score. The reported results of each method is averaged over trained models with 5 random seeds.

**Implementation Details.** For implementing the baselines, we use the default hyperparameters suggested in their respective papers and further fine-tune them to the best of our ability. For our CBD method, the diffuser backbone is based on Diffbid (Guo et al., 2024). The number of diffusion steps is set to 100, the horizon length of historical states is 3, and the batch size is 512 with a total of 500 training epochs. For the backbone of the diffuser, we adopt a temporal U-Net (Ronneberger et al.,

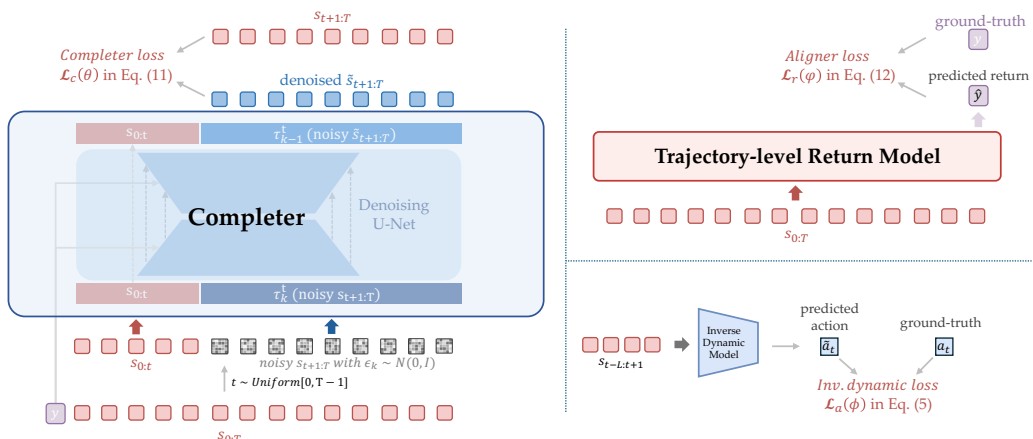

Figure 6: Demonstration of the training losses for each modules of our CBD method.

2015) with hidden sizes of [128, 256, 512]. The inverse dynamic model is implemented as an MLP with hidden sizes of [1024, 1024, 512]. The gradient guidance scale $\lambda$ is set as 0.1 following the previous typical setting of diffusers (Janner et al., 2022b) for all tasks. We train the model using the Adam optimizer (Loshchilov and Hutter) with a learning rate of 1e-4 on an H100 GPU. In this work, we adopt the setting from the previous decision diffuser (Ajay et al., 2023), where $y(\tau)$ is a scalar. In the main results shown in Tables 1 and 2, $y(\tau)$ is defined as $R(\tau) = \sum_{i=0}^{T} r_t$, which is scaled to $R(\tau) \in [0, 1]$ during training. During inference, $y(\tau) = 1$ is used to sample a high-return trajectory. In the general max conversion bidding setting, this approach is reasonable, as achieving more conversions implies utilizing more budget while maintaining a lower CPA.

**Details of Extension to Offline RL Tasks.** The offline reinforcement learning (RL) tasks under evaluation include three widely recognized locomotion challenges: HalfCheetah, Hopper, and Walker2d. These tasks involve maneuvering three distinct Mujoco robots to achieve optimal speed while ensuring energy efficiency and maintaining stability. The D4RL (Fu et al., 2020) benchmark offers offline datasets at three quality tiers: "medium," which includes demonstrations of moderate proficiency; "medium-replay," which encompasses all replay buffer recordings observed during training up to the point where the policy attains medium-level performance; and "medium-expert," which equally blends medium and expert-level performances. We also compared some additional non-diffuser-based methods on this benchmark, with details as follows: MOReL (Kidambi et al., 2020) addresses common pitfalls of model-based reinforcement learning, such as model exploitation, by learning a pessimistic Markov Decision Process (P-MDP) and training a near-optimal policy within this P-MDP. On the other hand, TT (Janner et al., 2021) employs a Transformer architecture to model distributions over trajectories and utilizes beam search as a planning algorithm.

## D  ALGORITHM

To provide a comprehensive understanding of our CBD method, we present a detailed algorithm outlining both the training and inference procedures in this section, as shown in Algorithm 1 and 2.

To offer a higher-level overview of the proposed method, we present a demonstration of the inference procedure in Figure 1 of the manuscript and a demonstration of the training procedure in Figure 6.

## E  ROBUSTNESS OF THE ALIGNER

For the principle of choosing a proper step size, as the offline data collected from online systems can be assumed to have a similar distribution to the online data, it is straightforward to use a portion of the offline data to evaluate the validity of different step sizes and employ the trajectory-level return model to assess alignment improvements. In this work, we choose the gradient step size of the aligner as 0.1, which generally brings improvement on various tasks. Furthermore, we have

---

**Algorithm 1 Training Procedure of CBD**

1: Initialize model parameters of completer $\epsilon_\theta$, aligner $R_\varphi$, and inverse dynamic model $f_\phi$
2: **for** each iteration **do**
3:     Sample $k \sim [1, K]$, $x_0 \sim \mathcal{D}$, $t \sim U\{0, ..., T-1\}$, $\boldsymbol{\epsilon} \sim \mathcal{N}(\mathbf{0}, \mathbf{I})$
4:     Compute the losses:
5:         $\mathcal{L}_c(\theta) = \mathbb{E}||\boldsymbol{\epsilon} - \epsilon_\theta(\tilde{\boldsymbol{x}}_k(\tau, t), k, \boldsymbol{y}(\tau))||^2_{t+1:T}$
6:         $\mathcal{L}_r(\varphi) = ||R_\varphi(\boldsymbol{x}_0(\tau)) - \boldsymbol{y}(\tau)||^2$
7:         $\mathcal{L}_a(\phi) = ||\boldsymbol{a}_t - f_\phi(\boldsymbol{s}_{t-L:t}, \boldsymbol{s}_{t+1})||^2$
8:     Update $\theta, \varphi, \phi$ using gradient descent on $\mathcal{L}_c(\theta), \mathcal{L}_r(\varphi)$, and $\mathcal{L}_a(\phi)$, respectively
9: **end for**

---

**Algorithm 2 Inference Procedure of CBD**

1: Initialize: initial state $\{\boldsymbol{s}_0\}$, completer $\epsilon_\theta$, aligner $R_\varphi$, inverse dynamic model $f_\phi$
2: **for** bidding timestep $t = 0$ to $T - 1$ **do**
3:     Generate future states $\{\tilde{\boldsymbol{s}}_{t+1}, ..., \tilde{\boldsymbol{s}}_T\}$ using completer $\epsilon_\theta$ by Eq. 8
4:     Do refinement by aligner: $\tilde{\boldsymbol{s}}'_{t+1} \leftarrow \tilde{\boldsymbol{s}}_{t+1:T} - \lambda\nabla_{\tilde{\boldsymbol{s}}_{t+1:T}}||R_\varphi(\tilde{\boldsymbol{x}}_0(\tau, t))||^2$
5:     Get action: $\boldsymbol{a}_t = f_\varphi(\boldsymbol{s}_{t-L:t}, \tilde{\boldsymbol{s}}'_{t+1})$
6:     Execute bids and get next state $\boldsymbol{s}_{t+1}$ from the ad system
7:     Update query: $\{\boldsymbol{s}_0, ..., \boldsymbol{s}_t, \boldsymbol{s}_{t+1}\} \leftarrow \{\boldsymbol{s}_0, ..., \boldsymbol{s}_t\} \oplus \boldsymbol{s}_{t+1}$
8: **end for**

---

included experimental results demonstrating the robustness of the gradient step size in Table 5. The validity is measured by the generated remaining budget's legitimacy. The GS-Align is an alternative alignment technique without gradient refinement, which is based on greedy selection of the best-aligned generated trajectory among parallelly sampled 10 generations, *i.e.*, best-of-N.

Table 5: Robustness and efficiency of the aligner's gradient step size.

| AuctionNet-Sparse | DiffBid | $\lambda = 0.001$ | $\lambda = 0.01$ | $\lambda = 0.05$ | $\lambda = 0.1$ | $\lambda = 1.0$ | $\lambda = 5.0$ | GS-Align |
|---|---|---|---|---|---|---|---|---|
| Value $\uparrow$ | 31.3 | 40.4 | 40.4 | 40.6 | 40.7 | 40.6 | 39.4 | 40.6 |
| ER $\downarrow$ | 1.23 | 0.91 | 0.91 | 0.87 | 0.86 | 0.91 | 0.87 | 0.88 |
| Score $\uparrow$ | 23.1 | 35.5 | 35.5 | 36.2 | 37.0 | 35.7 | 34.2 | 35.8 |
| Validity $\uparrow$ | 77.4% | 98.7% | 98.7% | 98.7% | 98.7% | 98.7% | 96.0% | 98.7% |

## F NECESSITY OF INVERSE DYNAMIC MODEL.

Current diffuser-based methods, such as Diffbid and our CBD method, generate states first and then derive an action, which is not straightforward since our major interest is in the action itself. This raises the question of whether we can directly generate the actions together with states, thereby avoiding the need for an additional inverse dynamic model. However, as shown in Table 2, our method still outperforms this variant (**CBD-$\mathcal{A}$**), empirically demonstrating the effectiveness and stability of training the model to learn planning first, resembling a chain-of-thought process (Wei et al., 2022) before determining the final action.

## G EXTENSION TO OFFLINE RL TASKS

Our CBD method could also bearing insights and benefit for general offline RL tasks like the locomotion tasks. We compare to the existing non-diffuser and diffuser-based methods together in Table 6, and a detailed introduction to these baselines can be seen in Appendix C. In some offline RL tasks, previous methods always set the first position of inputs as $\boldsymbol{s}_t$ and let the diffuser generate the remaining sequence (Dong et al., 2024b) and we also follow this setting in these tasks. However, we emphasize that this is not necessary for auto-bidding since the trajectory usually has a fixed length within a given period, *e.g.*, 48 steps in a day with a bidding frequency of every 30 minutes. Additionally, historical information plays a crucial role in bid assessment (Guo et al., 2024). As shown in Table 6, the CBD method achieves state-of-the-art performance, demonstrating its effectiveness and generalizations in offline RL tasks. Notably, our method performs slightly better in the *Medium-Expert* setting than in others, suggesting that transition quality also impacts performance, *i.e.*, the *Medium-Expert* data trains the model to generate both dynamic legitimate and expert-level transitions, whereas other settings only teach the model about dynamic legitimacy. This situation differs from auto-bidding tasks, where the offline training log is derived from real advertisers' behaviors, which inherently have a basic quality guarantee. Poor strategies will be immediately discontinued to prevent significant economic losses. However, this also presents unique challenges in developing new methods that can outperform existing competitive strategies in large-scale auctions.

Table 6: Performance comparison across different datasets and environments of the D4RL benchmark. Results correspond to the mean and standard error over 150 episode seeds, and the best scores of diffuser-based methods are emphasized in **bold**.

| Dataset | Environment | non-Diffuser-based | | | | | | Diffuser-based | | | | |
|---|---|---|---|---|---|---|---|---|---|---|---|---|
| | | BC | CQL | IQL | DT | TT | MOReL | Diffuser | DD | AD | DL | CBD |
| Medium-Expert | HalfCheetah | 55.2 | 91.6 | 86.7 | 86.8 | 95 | 53.3 | 79.8 | 90.6 | 90.4 | 88.5 | **93.5±0.83** |
| | Hopper | 52.5 | 105.4 | 91.5 | 107.6 | 110.0 | 108.7 | 107.2 | 111.8 | 109.3 | 111.6 | **113.8±0.13** |
| | Walker2d | 107.5 | 108.8 | 109.6 | 108.1 | 101.9 | 95.6 | 108.4 | 108.8 | 108.2 | 107.1 | **109.1±0.33** |
| Medium | HalfCheetah | 42.6 | 44.0 | 47.4 | 42.6 | 46.9 | 42.1 | 44.2 | 49.1 | 44.3 | 48.9 | **49.4±0.20** |
| | Hopper | 52.9 | 58.5 | 66.3 | 67.6 | 61.1 | 95.4 | 58.5 | 79.3 | 96.6 | **100.9** | 97.7±1.38 |
| | Walker2d | 75.3 | 72.5 | 78.3 | 74.0 | 79.0 | 77.8 | 79.7 | 82.5 | 84.4 | **88.8** | 83.0±0.42 |
| Medium-Replay | HalfCheetah | 36.6 | 45.5 | 44.2 | 36.6 | 41.9 | 40.2 | 42.2 | 39.3 | 38.3 | 41.6 | **42.8±0.31** |
| | Hopper | 18.1 | 95 | 94.7 | 82.7 | 91.5 | 93.6 | 96.8 | **100.0** | 92.2 | 96.6 | 96.5±0.18 |
| | Walker2d | 26.0 | 77.2 | 73.9 | 66.6 | 82.6 | 49.8 | 61.2 | 75.0 | 84.7 | **90.2** | 75.2±0.26 |

## H  ETHICS STATEMENT

To the best of our knowledge, there are no ethical issues with this paper.

## I  REPRODUCIBILITY STATEMENT

We provide the code attached in the supplementary materials to ensure reproducibility, with detailed implementation details in Appendix C.

## J  STATEMENT OF THE USE OF LARGE LANGUAGE MODELS

We use the LLMs to polish the writing by checking for grammatical errors.

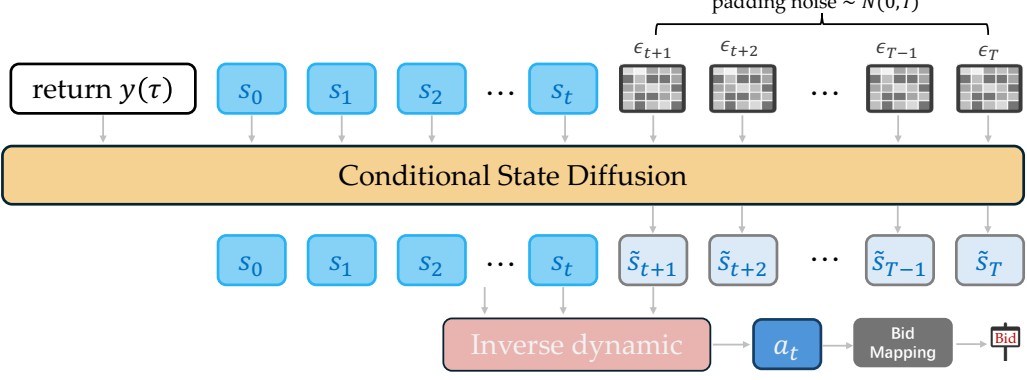

Figure 7: Overview of a typical diffuser-based decision-making procedure for auto-bidding: At each timestep for updating the bidding parameter, as introduced in Section 2.1, the diffuser first receives a trajectory consisting of a sequence of historical states $s_{0:t}$ and a sequence of padding noise $\epsilon_{t+1:T} \sim \mathcal{N}(\mathbf{0}, \boldsymbol{I})$, along with an expected return $\boldsymbol{y}(\tau)$ as the condition. The conditional state diffusion model then denoises the padding noise to generations $\tilde{\boldsymbol{s}}_{t+1:T}$. An inverse dynamic model projects the generated $\tilde{\boldsymbol{s}}_{t+1}$, together with $\boldsymbol{s}_{t-L:t}$, to the corresponding action $a_t$, which is the bidding parameter. During the extended period before the next decision-making timestep of bidding parameter update, which could last for a significant duration, such as half an hour, the bidding parameter $\boldsymbol{a}_t$ is continuously projected to the final bid for each impression opportunity using the bid mapping function in Eq. (2).

## K    MORE VISUALIZATION OF GENERATED TRAJECTORIES

We show more generated trajectories for visualization of the generation uncertainty issue in Fig. 8 and Fig. 9. The blue lines represent the actual trajectories found in the dataset, while the yellow lines depict the generated trajectories, which incorporate observations from the trajectory spanning from $t = 0$ to 23. A red line highlights the specific timestep at which decision-making occurs. As the results demonstrate, our CBD method effectively alleviates the issue of generation uncertainty by producing dynamic legitimate and well-aligned generations.

## L    BACKGROUND OF DIFFUSER-BASED DECISION-MAKING FOR AUTO-BIDDING

To facilitate understanding of the diffuser-based decision-making process for auto-bidding, we include an illustrative figure in Fig. 7, which is based on a commonly used decision diffuser (Ajay et al., 2023). The process consists of two main components. First, the decision diffuser takes in historical states and a return condition to generate future states. The action, specifically the bidding parameter $\lambda$ as defined in Eq. (2), is then derived using an inverse dynamic model. The latency discussed in Section 4.4 refers to this component. Second, once the action $a_t$ is determined at timestep $t$, it is continuously applied to the bidding of each impression opportunity until the next timestep $t + 1$ when the bidding parameters are needed to be updated. This application phase can extend over a longer duration, potentially lasting up to half an hour.

## M    DISCUSSION ON AUTO-REGRESSIVE AND MASKED TRAJECTORY MODELING METHODS

To distinguish our proposed method from prior auto-regressive and masked trajectory modeling approaches, we have included a detailed discussion in this section.

Auto-regressive methods, such as the Decision Transformer (DT), generate a single action using a causal transformer $p_\theta(\boldsymbol{a}_t | \boldsymbol{Rtg}_{\leq t}, \boldsymbol{s}_{\leq t}, \boldsymbol{a}_{<t})$, where $\boldsymbol{Rtg}_t$ represents the *return-to-go*. They are not directly modeling Eq. (10), *i.e.*, $p_\theta(\bar{\boldsymbol{s}}_{t+1:T} | \boldsymbol{s}_{\leq t}, \boldsymbol{R})$. In contrast, our CBD method, which falls under the category of diffusers, generates all future states in **one step** rather than using an auto-regressive approach, which is the key distinction from DT. Additionally, CBD can effectively leverages historical observations $s_{\leq t}$ to generate future states $s_{t+1:T}$, due to our augmented training process that learns to complete sequences of random lengths. A significant challenge in applying dynamic programming methods like DT to auto-bidding is its inability to handle sparse-reward settings. In DT, actions $\boldsymbol{a}_t$ are learned through the step-wise reward signals $r_t(\boldsymbol{s}_t, \boldsymbol{a}_t)$. However, $r_t$ could always be zero in the sparse training set. A zero reward could indicate either a poor action or a good action that did not receive a reward due to conversion randomness, making it difficult to accurately assess the value of single steps. Our method overcomes this limitation by conditioning on a trajectory-level accumulative rewards, alleviating the need for accurate single-step rewards. This advantage is supported by previous works (Ajay et al., 2023; Janner et al., 2022a) and our experiments in Table 1, where our method significantly outperforms DT in sparse settings. Although one could adapt DT to function as a "one-step" generator by auto-regressively generating states until the end of the trajectory, essentially implementing CBD with a decision transformer, we conducted experiments labeled "**CBD-Transformer**" in Table 2 and the experiment result of CBD-Transformer showed poor performance, underscoring the necessity of using a diffuser to achieve Eq. (10).

Masked trajectory modeling (MTM) methods (Wu et al., 2023; Zhang et al., 2025) are designed to mask random positions within a trajectory and train a model to reconstruct the original trajectory using a transformer. The primary goal is to enhance the representation learning of a backbone model, which can be applied to a variety of downstream tasks such as dynamic modeling, imitation learning, and action generation. Applying this MTM approach to diffusers based on transformers can be seen as a variant of DiT-causal. Recall that DiT-causal enforces the generation of next states strictly based on historical states for each timestep during training, accomplished using the masking mechanism

Table 7: Performance comparison between Completer and Completer (MTM) under Dense and Sparse version of AuctionNet benchmarks.

| Method | Dense | | | Sparse | | |
|---|---|---|---|---|---|---|
| | Value ↑ | ER ↓ | Score ↑ | Value ↑ | ER ↓ | Score ↑ |
| Completer | 370 | 1.12 | 292 | 40.4 | 0.90 | 35.5 |
| Completer (MTM) | 323 | 1.25 | 280 | 32.5 | 1.24 | 29.5 |

of causal attention. However, we found that DiT-causal performs poorly in experiments compared to our diffusion completer of CBD, as shown in Table 1 and Table 2. Intuitively, this could be due to the fact that a generated $\tilde{s}_{t+1}$ in CBD not only receives information from real historical states $s_{\leq t}$ but also from the noisy future generated $\tilde{s}^k_{>t+1}$ during the denoising steps $k = \{T, \ldots, 1\}$ in training and inference, forming a consistent trajectory based on a temporal U-Net backbone, which is a widely used and effective backbone in diffusers (Dong et al., 2024b). In contrast, DiT-causal strictly abandons the information from noisy generated $\tilde{s}^k_{>t+1}$ due to the masking mechanism, limiting its dynamic legitimacy. Therefore, instead of implementing MTM in transformers, we implemented MTM based on the same U-Net backbone as our CBD method. The masking ratio of MTM is set to 50% to keep a fair comparison with our Completer, which can observe an average of half the length of the trajectory. However, as shown in Table 7, results indicate that MTM could still limit the auto-bidding performance of diffusers. This could be due to the core motivation of this paper: the inconsistency between training and inference can affect performance. When applying MTM to a diffuser, it is trained to reconstruct a masked state $\tilde{s}_t$ by partially utilizing information from **incomplete** non-noisy historical states $masked(s_{<t})$. However, this is inconsistent with inference, where complete non-noisy historical states are available.

# N    EXPERIMENTS ON CLASSIFIER-GUIDANCE GENERATION

In the very beginning, the diffuser was implemented as a classifier-guidance scheme. Here, a trajectory-level return model $R(\tau)$ acts as the classifier providing gradient guidance during the denoising steps, expressed as $x_{k-1} \sim \mathcal{N}(\mu_k + \alpha \nabla_\theta \mathcal{J}(\mu_k), \sigma_k^2 I)$ (Janner et al., 2022a). However, this method has a severe drawback: The return model $R(\tau)$ is independently trained using real **non-noisy** trajectories and thus has not seen noisy trajectories during the denoising steps. These can be seen as out-of-distribution (OOD) cases, leading to gradient guidance failure. Unlike categorical distribution in image classifiers, the trajectory return distribution can be complex, and offline RL tasks often suffer from OOD errors (Kostrikov et al., 2022). In contrast, we apply the return model only in the final generation output, which is in the non-noisy trajectory distribution, exploiting the generative model's ability in generating in-distribution data. To support this claim, we provide the best achieved experimental results of adopting an classifer-guidance (CG) generation scheme as below, noted as "CBD(CG)".

Table 8: Performance comparison between CBD and CBD (CG) under Dense and Sparse version of AuctionNet benchmarks.

| Method | Dense | | | Sparse | | |
|---|---|---|---|---|---|---|
| | Value ↑ | ER ↓ | Score ↑ | Value ↑ | ER ↓ | Score ↑ |
| CBD | 374 | 1.10 | 298 | 40.7 | 0.86 | 37.0 |
| CBD (CG) | 350 | 1.15 | 290 | 32.2 | 1.23 | 29.7 |

# O    ERROR BAR

For a clearer demonstration of each method's robustness, Table 9 displays error bars calculated from 5 random seeds, corresponding to the main results in Table 1.

Table 9: Performance comparison with error bar across various methods under different budget conditions for AuctionNet and AuctionNet-sparse datasets.

| Dataset | Budget | RL | | | | Transformer | | | Diffuser | | | | | Improve |
|---|---|---|---|---|---|---|---|---|---|---|---|---|---|---|
| | | USCB | BCQ | CQL | IQL | DT | CDT | DT-S | DiffBid | DiT | DiT-causal | CBD-Completer | CBD | |
| **Dense** | 50% | 133.1±3.9 | 184.2±5.3 | 180.1±5.6 | 185.0±5.2 | 186.2±1.9 | 190.3±1.8 | 196.8±2.1 | 161.9±3.1 | 156.4±3.3 | 160.0±3.0 | 191.9±2.0 | **198.6±1.8** | +22.3% |
| | 75% | 201.9±5.8 | 260.1±7.5 | 256.6±8.0 | 259.9±7.3 | 234.1±2.2 | 290.8±2.7 | 298.0±3.1 | 233.7±4.9 | 228.3±4.3 | 229.1±4.7 | 280.3±3.4 | **298.3±2.6** | +27.8% |
| | 100% | 267.4±7.6 | 270.6±8.4 | 336.6±9.8 | 322.2±9.1 | 364.0±3.5 | 366.5±3.8 | 373.1±3.6 | 316.5±6.5 | 307.0±5.9 | 311.8±6.1 | 370.4±3.2 | **374.0±3.0** | +18.3% |
| | 125% | 335.0±9.7 | 333.0±10.3 | 400.2±11.6 | 398.6±11.2 | 393.0±4.1 | 400.2±3.7 | 418.6±4.3 | 384.0±7.9 | 375.1±7.2 | 381.8±7.5 | 420.5±4.1 | **427.2±4.0** | +11.1% |
| | 150% | 415.9±12.1 | 388.5±12.0 | 465.3±13.5 | 457.3±13.9 | 445.2±4.3 | 429.5±4.5 | 437.6±4.6 | 452.5±8.8 | 435.1±9.0 | 440.6±8.5 | 473.7±5.2 | **480.5±4.6** | +6.19% |
| **Sparse** | 50% | 15.42±0.45 | 18.10±0.52 | 16.79±0.52 | 20.21±0.58 | 18.20±0.19 | 18.01±0.17 | 19.62±0.21 | 14.54±0.30 | 14.12±0.27 | 14.33±0.29 | 20.08±0.29 | **20.56±0.21** | +41.4% |
| | 75% | 21.63±0.63 | 27.00±0.78 | 21.80±0.68 | 26.90±0.77 | 27.50±0.26 | 27.54±0.29 | 28.70±0.30 | 23.25±0.48 | 22.58±0.43 | 22.60±0.47 | 29.90±0.28 | **29.97±0.26** | +28.9% |
| | 100% | 28.48±0.83 | 30.54±0.94 | 30.15±0.88 | 36.06±1.05 | 31.30±0.33 | 34.31±0.32 | 36.82±0.35 | 31.33±0.64 | 31.02±0.60 | 31.27±0.65 | 40.40±0.38 | **40.71±0.31** | +29.9% |
| | 125% | 34.88±1.01 | 35.19±1.09 | 37.67±1.10 | 43.92±1.27 | 43.51±0.42 | 42.52±0.44 | 43.91±0.45 | 40.54±0.79 | 39.40±0.81 | 39.14±0.76 | 45.54±0.48 | **46.04±0.45** | +13.5% |
| | 150% | 43.08±1.25 | 36.97±1.14 | 44.43±1.29 | 49.27±1.43 | 50.04±0.52 | 50.93±0.49 | 49.65±0.51 | 47.18±0.92 | 45.81±0.94 | 46.07±0.90 | 51.10±0.49 | **51.35±0.47** | +8.91% |

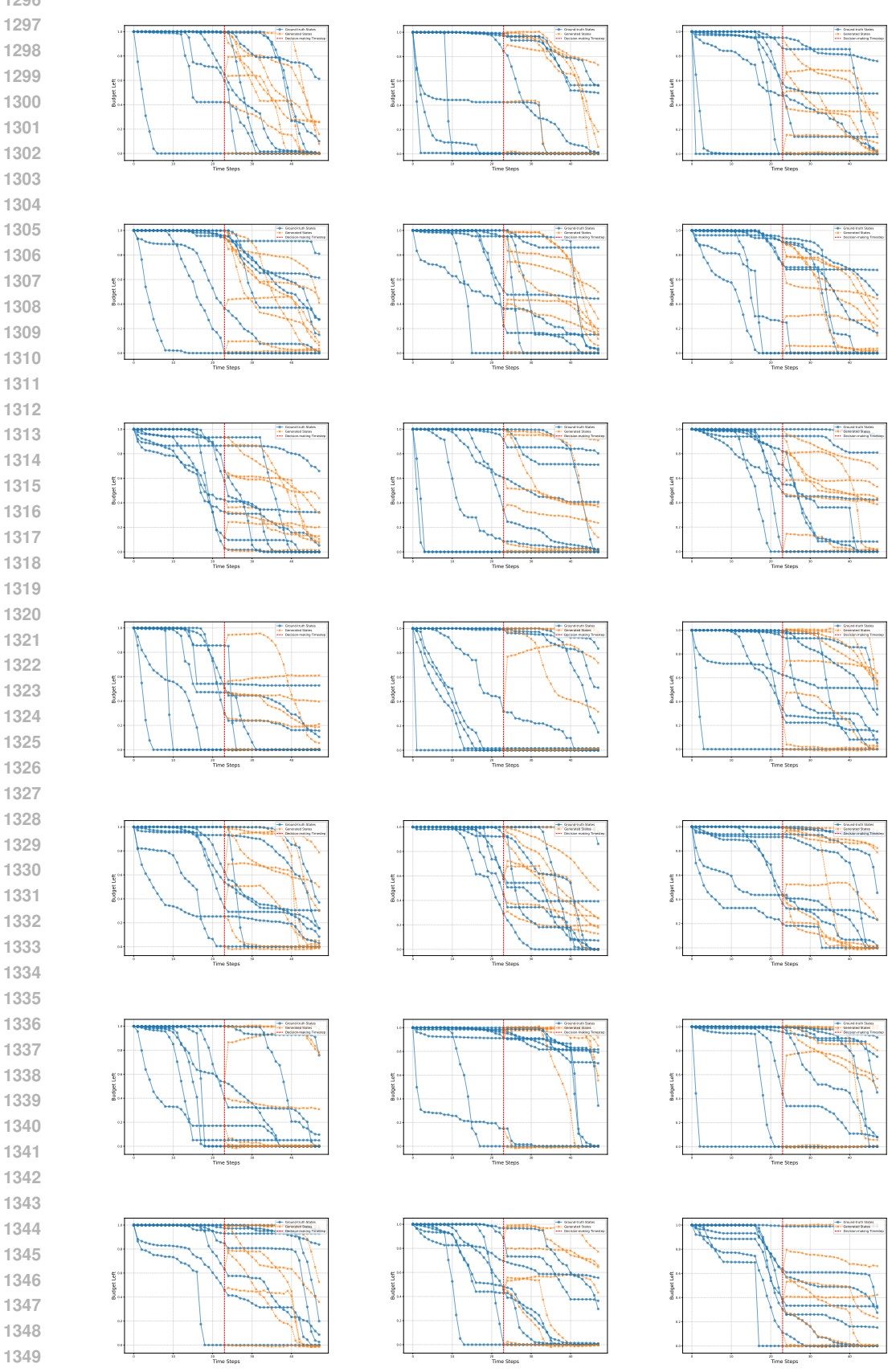

Figure 8: Generated trajectories of DiffBid.

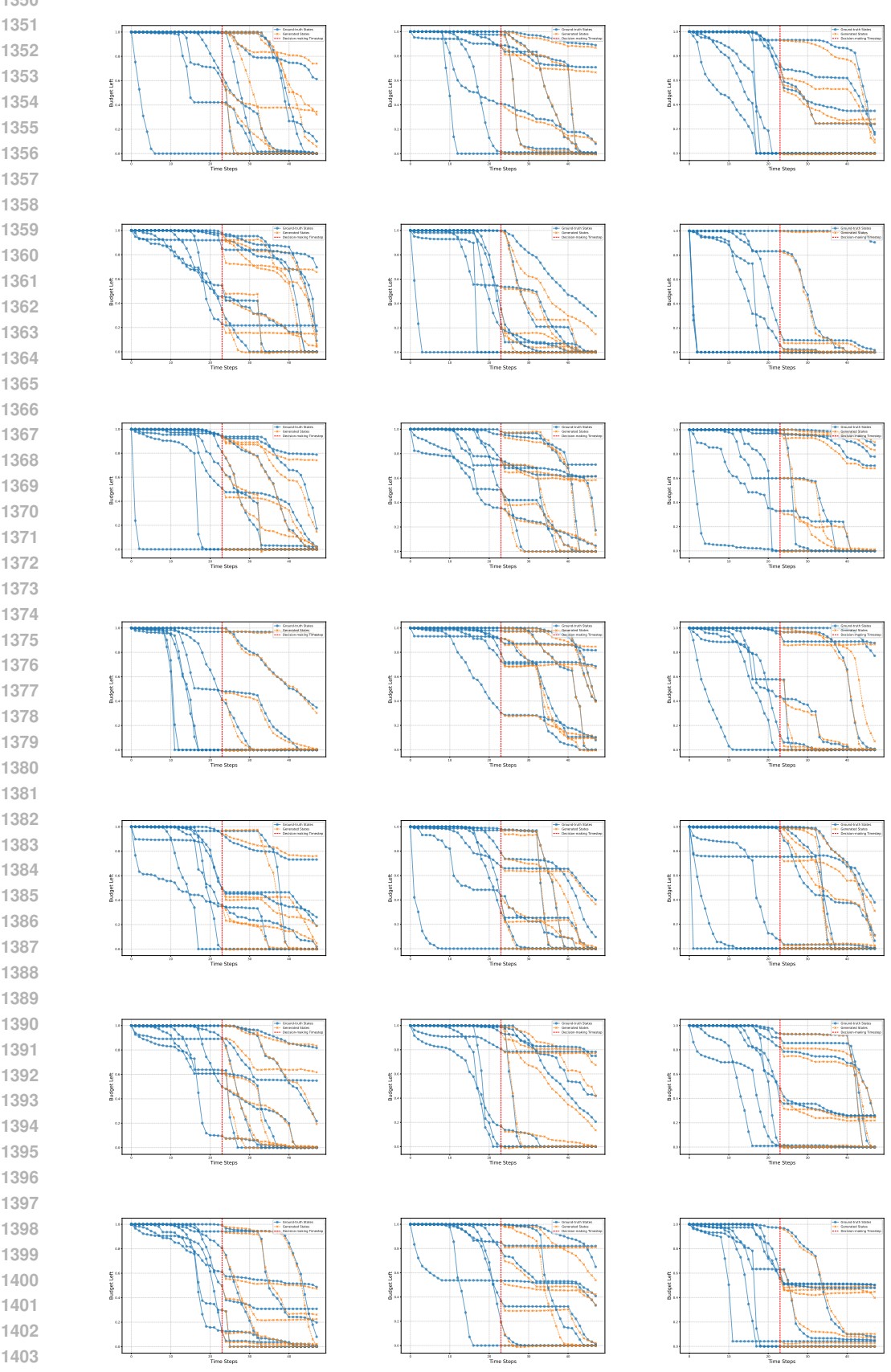

Figure 9: Generated trajectories of CBD.

