# OpenReview forum: "Generative Auto-Bidding in Large-Scale Auctions via Diffusion Completer-Aligner"
_ICLR.cc/2026/Conference — Submitted to ICLR 2026_

### Official Review · Reviewer_cGA9 · 2025-10-22

**Soundness:** 3
**Presentation:** 1
**Contribution:** 3
**Rating:** 8
**Confidence:** 3

**Summary:**

The paper proposes a training procedure to improve diffusion models that participate in auto-bidding systems by introducing completer and aligner components. The diffuser's objective is producing bidding parameters, meaning it does not have to run quickly within milliseconds per ad request, and thus can be a fairly complex model. The additional two components aim to improve the legitimacy of the taken actions and alignment with cumulative reward of the campaign being optimized.

**Strengths:**

1. Paper clearly explains the motivation behind building a complex and expensive model for an apparently real-time task.
2. The idea of learning to complete appears to be simple and creative.
3. Performance improvement against baselines is apparent. I do not believe additional datasets should be used, since there aren't any around for simulating actual auctions that I know about (with actual competitors, where the model under test has to participate in auctions, rather than just predict some value).
4. Pretty extensive ablation studies.

**Weaknesses:**

1. The basics of diffusion-based decision making are very poorly explained, and pretty much assume the reader is familiar. So either you assume the reader is familiar, and just refer to references, or assume the reader is not, and explain clearly what is it. I am not familiar with the idea, and had to go through relevant literature to understand the explanation, and then it was actually redundant when I understood it, and could be shorter. Maybe a simple figure instead of a lot of words and jargon could be better here.
2. Auto-bidding could be many things, such as bid shading, for example. The paper's abstract and title need to be more precise in setting the stage.
3. One thing is not clear - it appears as if the diffusion model is supposed to generate the lambda parameters that are used for bidding, but then in the A/B test section, latency is reported. Shouldn't the latency cost be zero?

**Questions:**

1. Why is the latency reported, if as far as I understand your proposed model doesn't even compute the auction bids, but just the lambdas that are used for bidding? Please clarify. I assume I missed an important detail.

---

> ### Author Response · Authors · 2025-11-20
> **Response to reviewer cGA9**
>
> Thank you very much for your recognition and we hope our response could clarify your confusions.
>
> ---
> **Weakness 1：illustrative figure of diffuser-based decision-making**
>
> We deeply appreciate this suggestion, and we have added an illustrative figure of diffuser-based decision-making in **Figure 7 of Appendix L**.
>
> ---
> **weakness 2: scope of the work**
>
> We greatly appreciate this suggestion. We agree that auto-bidding can encompass a wide range of research directions, such as auction mechanism design and bid optimization strategies, particularly the bid shading you mentioned, which focuses on predicting the second-highest market price and is applied in Generalized First Price auction mechanisms.
> Our bid optimization is formulated as a sequential decision-making process that dynamically updates the bidding parameters to maximize the
> total accumulated impression value for an advertiser while adhering to some economic constraints.
> We have revised the abstract to more clearly reflect that the scope of this work is **bid optimization strategy** with sequential decision-making techniques.
> The title will be revised as "Generative Auto-Bidding Strategy in Large-Scale Auctions via Diffusion Completer-Aligner".
>
> ---
> **weakness 3 & Question 1: inference procedure**
>
> We sincerely apologize for any confusion caused. Your understanding is correct regarding the computation of auction bids, which is performed using Eq. (2) with zero latency, utilizing lambda parameters generated by an auto-bidding model.
> The reported latency in an online A/B test is measured from the moment a bidding parameter adjustment service request is received until the updated bidding parameter (i.e., the lambda parameters) is returned.
> As a result, there will be some latency associated with the model inference.
>
> Once the bidding parameters are updated, it will take effect for a longer period until the next adjustment requirement comes, as introduced in Section 2.1 and Appendix L.
>
> ----
> Thanks again for your insightful comments! We would be delighted to engage in further discussion with you during the rebuttal stage.

---

> > ### Comment · Reviewer_cGA9 · 2025-11-22
> > **Keep score**
> >
> > Thank you for tve clarifications. I will maintain my (positive) score of recommending to accept the paper, since the weaknesses are mostly of technical nature.

---

> > > ### Author Response · Authors · 2025-11-25
> > >
> > > Dear Reviewer cGA9,
> > >
> > > Thank you so much for your feedback, and we are very pleased to have addressed your concerns! We want to express our deepest gratitude for your recognition and support of our work!
> > >
> > > Warm regards,
> > >
> > > The Authors

---

### Official Review · Reviewer_2xeh · 2025-10-26

**Soundness:** 2
**Presentation:** 3
**Contribution:** 2
**Rating:** 4
**Confidence:** 3

**Summary:**

In this paper, the authors propose a novel methodology called CBD (Causal auto-bidding method based on a diffusion completer-aligner framework) for addressing the auto-bidding problem in the advertising domain. The proposed approach aims to mitigate the uncertainty inherent in generative models and the paper also provides a theoretical analysis supporting this aspect.
In the experimental section, the authors evaluate the performance of the proposed methodology under different budget constraints and against several baselines. The results show that the proposed method consistently outperforms the compared approaches, achieving particularly higher improvements in low-budget scenarios. Moreover, as reported in Section 5, the authors deployed the proposed algorithm on an online advertising platform, where it improved both the Target Cost and CPA Valid Ratio without increasing overall costs—demonstrating the practical applicability of the proposed methodology.

**Strengths:**

- The paper is complete and well written
- The proposed methodology is well justified well benchmarked against several baselines
- The test conducted on a online advertising platform is a good point, demostrating the practical usability of the proposed methodology

**Weaknesses:**

- While Table 6 reports the performance of CBD on the D4RL benchmark using multiple seeds and standard errors, Table 1 presents metrics without confidence intervals or measures of variability.
- As acknowledged by the authors, a key limitation of the work is its exclusive focus on the auto-bidding strategy, without addressing other important aspects such as auction mechanism design or cooperation between advertisers.

Overall, while the proposed methodology appears promising, I have some concerns regarding the evaluation process, particularly about the consistency and robustness of the reported results.

**Questions:**

- Could you clarify why the performance reported in Table 1 is presented as a single value?
- More generally, did you consider multiple random seeds in your experiments? Aside from the D4RL Benchmark, I couldn’t find any other mention of this detail in the paper.

---

> ### Author Response · Authors · 2025-11-20
> **Response to reviewer 2xeh**
>
> Thank you for your tremendous effort and helpful comments in reviewing our work!
>
> ---
> **Weakness 1& Questions: Consistency and Robustness**
>
> We deeply appreciate the comments and suggestions regarding robustness, and we are really sorry for the lack of experimental details. The evaluation procedure for the methods is strictly based on the AuctionNet Benchmark, and we report the results for each method averaged over 5 random seeds, as is common practice.
> Additionally, the improvements specifically compared to the backbone DiffBid are statistically significant as determined by a two-sided t-test with p<0.05. This is a common metric for robustness in computational advertising and recommendation systems.
> In addition to Table 1, the experiments in Table 5 of the Appendix further support the robustness of the method. We will add these details to the experimental setup section.
>
> To better illustrate the robustness of each method, we report the error bars for the main results in Table 1 as below (added to Table 9 of Appendix O):
>
> | Dataset | Budget | **RL** |  |  |  | **Transformer** |  |  | **Diffuser** |  |  |  |  | improve |
> |---------|--------|--------|--------|--------|--------|--------|--------|--------|--------|--------|--------|--------|--------|---------|
> |  |  | USCB | BCQ | CQL | IQL | DT | CDT | DT-S | DiffBid | DiT | DiT-causal | CBD-Completer | CBD |  |
> | **AuctionNet** | 50% | 133.1±3.9 | 184.2±5.3 | 180.1±5.6 | 185.0±5.2 | 186.2±1.9 | 190.3±1.8 | 196.8±2.1 | 161.9±3.1 | 156.4±3.3 | 160.0±3.0 | 191.9±2.0 | **198.6±1.8** | +22.3% |
> |  | 75% | 201.9±5.8 | 260.1±7.5 | 256.6±8.0 | 259.9±7.3 | 234.1±2.2 | 290.8±2.7 | 298.0±3.1 |233.7±4.9| 228.3±4.3 | 229.1±4.7 | 280.3±3.4 | **298.3±2.6** | +27.8% |
> |  | 100% | 267.4±7.6 | 270.6±8.4 | 336.6±9.8 | 322.2±9.1 | 364.0±3.5 | 366.5±3.8 | 373.1±3.6 |316.5±6.5| 307.0±5.9 | 311.8±6.1 | 370.4±3.2 | **374.0±3.0** | +18.3% |
> |  | 125% | 335.0±9.7 | 333.0±10.3 | 400.2±11.6 | 398.6±11.2 | 393.0±4.1 | 400.2±3.7 | 418.6±4.3 | 384.0±7.9 | 375.1±7.2 | 381.8±7.5 | 420.5±4.1 | **427.2±4.0** | +11.1% |
> |  | 150% | 415.9±12.1 | 388.5±12.0 | 465.3±13.5 | 457.3±13.9 | 445.2±4.3 | 429.5±4.5 | 437.6±4.6 | 452.5±8.8 | 435.1±9.0 | 440.6±8.5 | 473.7±5.2 | **480.5±4.6** | +6.19% |
> | **AuctionNet-sparse** | 50% | 15.42±0.45 | 18.10±0.52 | 16.79±0.52 | 20.21±0.58 | 18.20±0.19 | 18.01±0.17 | 19.62±0.21 | 14.54±0.30 | 14.12±0.27 | 14.33±0.29 | 20.08±0.29 | **20.56±0.21** | +41.4% |
> |  | 75% | 21.63±0.63 | 27.00±0.78 | 21.80±0.68 | 26.90±0.77 | 27.50±0.26 | 27.54±0.29 | 28.70±0.30 | 23.25±0.48 | 22.58±0.43 | 22.60±0.47 | 29.90±0.28 | **29.97±0.26** | +28.9% |
> |  | 100% | 28.48±0.83 | 30.54±0.94 | 30.15±0.88 | 36.06±1.05 | 31.30±0.33 | 34.31±0.32 | 36.82±0.35 | 31.33±0.64 | 31.02±0.60 | 31.27±0.65 | 40.40±0.38 | **40.71±0.31** | +29.9% |
> |  | 125% | 34.88±1.01 | 35.19±1.09 | 37.67±1.10 | 43.92±1.27 | 43.51±0.42 | 42.52±0.44 | 43.91±0.45 | 40.54±0.79 | 39.40±0.81 | 39.14±0.76 | 45.54±0.48 | **46.04±0.45** | +13.5% |
> |  | 150% | 43.08±1.25 | 36.97±1.14 | 44.43±1.29 | 49.27±1.43 | 50.04±0.52 | 50.93±0.49 | 49.65±0.51 |47.18±0.92 | 45.81±0.94 | 46.07±0.90 | 51.10±0.49 | **51.35±0.47** | +8.91% |
>
> ---
> **Weakness 2: Auction mechanism design and real-world complexity**
>
> Yes, the design of auction mechanisms and the complex relationships between advertisers are indeed fascinating topics. Analyzing their influence on auto-bidding strategies could be a meaningful direction for future work.
> Regarding auction mechanisms, they are not frequently updated in industrial ad platforms, where Generalized Second Price (GSP) auctions are widely applied. This broad application in the industry suggests that our method has a wide range of applicability.
> As for the complex relationships between advertisers and other real-world complexities, such as manual interventions like early stopping of auto-bidding services, replicating these components within a public benchmark is challenging. However, the effectiveness of CBD in handling these real-world complexities could be empirically supported by our online A/B test results.
>
> ---
> We deeply apologize for any confusion that may have arisen and hope our response has effectively addressed your concerns. Your feedback is very important to us, and we would be eager to discuss it further with you during the rebuttal stage.

---

> ### Author Response · Authors · 2025-11-25
>
> Dear Reviewer 2xeh,
>
> We are sincerely grateful for the time and effort you've spent reviewing our work. With the rebuttal period closing in a week, we hope our response has adequately addressed your concerns.
>
> In summary, we have provided a complete set of **error bars** for a clearer evaluation of the robustness of the CBD method, which notably achieves much better performance in sparse scenarios. The superior performance of our method in the online A/B test further supports its **effectiveness in addressing real-world complexities**.
>
> Your feedback is of great significance to us, and if our responses have met your expectations, we would be sincerely appreciative if you could consider raising the score. Should you have any further questions or concerns, we would be delighted to discuss them and provide any additional clarifications needed to ensure all your concerns are fully addressed.
>
> Warm regards,
>
> The Authors

---

> > ### Comment · Reviewer_2xeh · 2025-11-28
> >
> > Thanks for the answers. I will revise my rating accordingly.

---

> > > ### Author Response · Authors · 2025-11-28
> > >
> > > Dear Reviewer 2xeh,
> > >
> > > Thank you so much for your positive feedback! We are delighted to have addressed your concerns and sincerely appreciate your recognition and willingness to revise the rating.
> > >
> > > Warm regards,
> > >
> > > The Authors

---

### Official Review · Reviewer_GWa8 · 2025-10-31

**Soundness:** 3
**Presentation:** 3
**Contribution:** 2
**Rating:** 4
**Confidence:** 5

**Summary:**

This paper examines the application of diffusion-based generative models to industrial-scale auto-bidding in online advertising. The authors identify two key failure modes of standard diffusion planners, namely dynamic illegitimacy and preference misalignment; the former is a nuanced and interesting issue that requires careful analysis. To mitigate these failure modes, the paper proposes CBD (Causal Auto-Bidding via Diffusion Completer–Aligner), a two-stage framework consisting of a Completer and an Aligner. The Completer augments training with an additional random variable t to enhance dynamic legitimacy between adjacent states. The Aligner employs a trajectory-level return model to refine generated trajectories during inference, aligning them more closely with advertisers’ objectives. The method is evaluated on AuctionNet and its sparse variant, showing consistent gains over baselines (e.g., +29.9% conversion value on AuctionNet-sparse). An online A/B test reports a +2.0% increase in conversions with no budget increase.

**Strengths:**

1. The paper articulates a practical issue—dynamic illegitimacy—that requires careful scrutiny and analyzes its causes. It addresses dynamic illegitimacy in diffusion-based auto-bidding via a Completer–Aligner framework. The theoretical analysis of distribution mismatch (Eq. 6) provides insight into diffusion-model failures.
2. The paper presents extensive experiments on public benchmarks (AuctionNet) and in real-world deployment (online A/B tests). The improvement in sparse-reward settings is substantial.

**Weaknesses:**

1. Overstated novelty: The “Completer” is similar to masked-trajectory modeling (akin to Decision Diffuser with partial observations). While applying it to diffusion models is novel, the core idea resembles concurrent work. The paper does not adequately differentiate itself.
2. The resolution of dynamic illegitimacy is not reported in detail—for example, quantitative metrics before and after mitigation are missing. It is unclear whether the issue is fully resolved.

**Questions:**

1. What exactly is $y(\tau)$ in Eq. 3 and Eq. 10? Is it the total conversion value ($\sum_t r_t$), or a vector of KPIs (e.g., [conversions, CPA])? How is it computed in sparse-reward settings where most $r_t = 0$?
2. In Algorithm 2, Step 5 uses only the refined next state $ \tilde{s}'_ {t+1} $ (not the full refined suffix $ \tilde{s}'_ {t+1 : T} $) for action generation. Why not use more future states (e.g., $\tilde{s}'_{t+1:t+L}$) to improve action quality?
3. Aligner vs. classifier guidance: Have you compared your gradient-based Aligner with classifier-guided diffusion (using $R_\varphi$ as the classifier)? Classifier guidance operates during denoising and may yield better alignment with similar computational cost.
4. You mention only a 6 ms overhead relative to DT, yet the paper uses 100 diffusion steps, which is theoretically costly. Could you provide details of the online inference setup (e.g., number of diffusion steps, batching, precision) and the corresponding GPU configuration?

---

> ### Author Response · Authors · 2025-11-20
> **Response to reviewer GWa8 (1/2)**
>
> We greatly appreciate your significant effort and the invaluable insights you provided in reviewing our work!
>
> ---
> **Weakness 1: difference to masked trajectory modeling**
>
> Thanks for your helpful suggestion! To differentiate our proposed method from concurrent related works, we have added a detailed discussion in **Appendix M** as follows:
>
> Masked trajectory modeling (MTM) methods [1,2] are designed to mask random positions within a trajectory and train a model to reconstruct the original trajectory using a transformer. The primary goal is to enhance the representation learning of a backbone model, which can be applied to a variety of downstream tasks such as dynamic modeling, imitation learning, and action generation.
> Applying this MTM approach to diffusers based on transformers can be seen as a variant of DiT-causal. Recall that DiT-causal enforces the generation of next states strictly based on historical states for each timestep during training, accomplished using the masking mechanism of causal attention. However, we found that DiT-causal performs poorly in experiments compared to our diffusion completer of CBD, as shown in Tables 1 & 2.
> Intuitively, this could be due to the fact that a generated $\tilde{s}\_{t+1}$ in CBD not only receives information from real historical states $s_{\leq t}$ but also from the noisy future generated $\tilde{s}\_{>t+1}^k$ during the denoising steps $k=\{T,\ldots,1\}$ in training and inference, forming a consistent trajectory based on a temporal U-Net backbone, which is a widely used and effective backbone in diffusers [3]. In contrast, DiT-causal strictly abandons the information from noisy generated $\tilde{s}\_{>t+1}^k$ due to the masking mechanism, limiting its dynamic legitimacy.
>
> Therefore, instead of implementing MTM in transformers, we tried to implement MTM based on the same U-Net backbone as our CBD method. The masking ratio of MTM is set to 50\% to keep a fair comparison with our Completer, which can observe an average of half the length of the trajectory.
> However, as shown in the Table below, results indicate that MTM could still limit the auto-bidding performance of diffusers. This could be due to the core motivation of this paper: the inconsistency between training and inference can affect performance. When applying MTM to a diffuser, it is trained to reconstruct a masked state $\tilde{{s}}\_{t}$ by partially utilizing information from the **incomplete** non-noisy historical states $\textit{masked}({s}\_{<t})$. However, this is inconsistent with inference, where complete non-noisy historical states are available.
>
>
> | Method | |Dense| | |Sparse| |
> | -------- | ---|--------|--- | -------- | ---| ---|
> |    | Value $\uparrow$ | ER $\downarrow$| Score $\uparrow$| Value $\uparrow$    | ER $\downarrow$   | Score $\uparrow$|
> |Completer    | 370  | 1.12  |  292  | 40.4    |  0.90      |  35.5 |
> |Completer(MTM)     |  323   | 1.25  |  280  | 32.5    |  1.24      | 29.5 |
>
>
> [1] Wu, Philipp, et al. "Masked trajectory models for prediction, representation, and control." ICML 2023.
>
> [2] Zhang, Xinzhi, et al. "Efficient Decision Sequence Modeling via Feature-Level Masking." 2025.
>
> [3] Dong, Zibin, et al. "Cleandiffuser: An easy-to-use modularized library for diffusion models in decision making." NeurIPS 2024.
>
>
>
>
>
> ---
> **Weakness 2: Dynamic legitimacy**
>
> For the quantitative evaluation of dynamic legitimacy, although it is challenging or even impossible to have a ground-truth forward dynamic model $p(s_{t+1}|s_t)$ for evaluation, we can empirically choose the element "remaining budget" in the state as a representative and essential indicator of dynamic legitimacy. Please refer to the "validity" results in Table 5 of Appendix E, which are measured by the generated remaining budget’s dynamic legitimacy.
> The diffbid baseline achieves only 77.4\% validity on the test set, while our method achieves 98.7\% validity of dynamic legitimacy. (The validity of the diffusion completer's generations corresponds to the GS-Align version, which does not perform gradient refinement on the generated trajectory; the validity after further gradient refinement by an aligner corresponds to the $\lambda=0.1$ setting.)
>
>
> ---
> **Q1: setup of $y(\tau)$**
>
> We sincerely apologize for this confusion.
> In this work, we adopt the setting from the previous decision diffuser [4], where $y(\tau)$ is a scalar. For instance,  in the main results shown in Tables 1 and 2, $y(\tau)$ is defined as $R(\tau)=\sum_{i=0}^{T} r_t$ as stated in line 177, which is scaled to $R(\tau)\in[0,1]$ during training. During inference, $y(\tau)=1$ is used to sample a high-return trajectory. In the general max conversion bidding setting, this approach is reasonable, as achieving more conversions implies utilizing more budget while maintaining a lower CPA.
> We have added these details to Appendix C.
>
> [4] Ajay, Anurag, et al. "Is conditional generative modeling all you need for decision-making?" ICLR 2023.

---

> > ### Author Response · Authors · 2025-11-20
> > **Response to reviewer GWa8 (2/2)**
> >
> > ---
> > **Q2: action generation**
> >
> > We greatly like this insightful comment about using more future states to enhance action quality! There are two reasons for our approach:
> > 1)  The inverse dynamic model receiving $\{s_{t-L:t}, \tilde{s}\_{t+1}\}$ is a common practice in previous diffuser methods.
> > This setting is practical and necessary for episodic decision-making tasks.
> > For example, when it comes to the final timestep $s_{T-1}$, we only have the generation $s_T$. In this case, the inverse dynamic model can only receive one future state for action prediction.
> >
> > 2) Unlike previous diffusers that typically abandon all future states $\tilde{s}\_{t+2:T}$ in action prediction with a inverse dynamic model, we use the aligner to exploit these future states $\tilde{s}\_{t+2:T}$ to enhance the quality of $\tilde{s}\_{t+1}$, which further influences the quality of action $a_t$.
> >
> >
> > ---
> > **Q3: classfier-guidance**
> >
> > We sincerely appreciate your thoughtful review of our work! Yes, actually, in the very beginning, the diffuser was implemented as a classifier-guidance scheme. Here, $R(\tau)$ acts as the classifier as you mentioned, providing gradient guidance during the denoising steps, expressed as $x_{k-1}\sim\mathcal{N}(\mu_k+\alpha\nabla_\theta\mathcal{J}(\mu_k), \sigma_k^2{I})$ [5], as discussed in line 304 of Section 3.2.
> >
> > However, this method has a severe drawback: The Return model is independently trained using real **non-noisy** trajectories and thus has not seen noisy trajectories during the denoising steps. These can be seen as out-of-distribution (OOD) cases, leading to gradient guidance failure. Unlike categorical distribution in image classifiers, the trajectory return distribution can be complex, and offline RL tasks often suffer from OOD errors [6]. In contrast, we apply the return model only in the final generation output, which is in the non-noisy trajectory distribution, exploiting the generative model's ability to generate in-distribution data.
> > To support this claim, we provide the best achieved experimental results of adopting a classifier-guidance (CG) generation scheme as below, noted as "CBD(CG)".
> > | Method | |Dense| | |Sparse| |
> > | -------- | ---|--------|--- | -------- | ---| ---|
> > |    | Value $\uparrow$ | ER $\downarrow$| Score $\uparrow$| Value $\uparrow$    | ER $\downarrow$   | Score $\uparrow$|
> > |CBD    | 374  | 1.10  |  298  | 40.7    |  0.86      |  37.0 |
> > |CBD(CG)     |  350   | 1.15  |  290  | 32.2    |  1.23      | 29.7 |
> >
> >
> > [5] Janner, Michael, et al. "Planning with diffusion for flexible behavior synthesis." ICML 2022.
> >
> > [6] Kostrikov, Ilya, Ashvin Nair, and Sergey Levine. "Offline reinforcement learning with implicit q-learning." ICLR 2022.
> >
> >
> > ---
> > **Q4: Online Setup**
> >
> > For the online inference setup, as we cannot modify the in-production method due to safety concerns, the diffuser is configured with similar parameters to the in-production DT-based method, totaling around 10M. We did not include DiffBid in our comparison due to its poor performance in offline experiments, which could lead to significant economic losses if deployed in an online system.
> > The denosing steps have been reduced to 10 for faster inference, which already significantly outperforms the DT-based method, showcasing the CBD's superiority in handling the sparse conversion settings typical of online advertising environments.
> >
> > In industrial auto-bidding services, ad platforms utilize engineering techniques like TorchScript and C++ to accelerate model services, offering much faster performance compared to directly using Python code. Both methods employed around 350 CPU cores for computation (we do not use GPUs currently), and inference latency was measured using these resources.
> > With the same budget allocated for each method’s experiment, each method achieved a daily average of approximately
> > 80,000,000 impressions and 14700 campaigns.
> > The average Queries Per Second (QPS, akin to the "batch" you mentioned) for service calls is 896, with our method achieving an average service availability of 99.8%, matching the DT method's availability.
> >
> >
> > ---
> > Thank you once again for your effort in reviewing our work! We sincerely apologize for any misunderstandings that may have arisen and hope that our response fully addresses your concerns. We would be more than happy to discuss this further with you during the rebuttal stage.

---

> ### Author Response · Authors · 2025-11-25
>
> Dear Reviewer GWa8,
>
> We greatly appreciate the dedication and time you've put into reviewing our work. As the rebuttal period is nearing its end in a week, we kindly inquire if our response has adequately addressed your concerns.
>
> To enhance our work, we have provided a detailed **discussion on the differences from existing works**, including masked trajectory modeling (MTM), and conducted an **additional experiment implementing CBD with MTM**. We have also highlighted the empirical **evaluation of dynamic legitimacy**. Our approach uses more generated states to improve **action quality** through an aligner in an indirect manner compared to previous diffusers. Additionally, we have included the necessary experiments with **classifier-guidance generation** and offered more details on both the offline and online **experimental settings**.
>
> Your feedback is crucial to us, and if our responses have satisfactorily addressed your concerns, we would be sincerely thankful if you could consider raising the score.  Should you have any further questions, we are more than willing to address them to ensure all your concerns are thoroughly resolved.
>
> Warm regards,
>
> The Authors

---

### Official Review · Reviewer_Fvh3 · 2025-11-01

**Soundness:** 3
**Presentation:** 2
**Contribution:** 3
**Rating:** 4
**Confidence:** 3

**Summary:**

The paper introduces a novel framework to improve auto-bidding in large-scale online advertising auctions. It introduces a Completer to reformulate difusser training as a completion process which augments the model with a random variable, and a Aligner to employ a trajectory-level return model to refine the generated trajectories. Comprehensive offline and online A/B test proves the effectiveness of the proposed method.

**Strengths:**

1. Motivation. The paper identifies generation uncertainty (illegitimacy and misalignment) as a key bottleneck in applying diffusion-based policies to industrial auto-bidding.

2. Evaluation. The authorsn conducted comprehensive offline evaluation and ablation study, as well as online A/B tests.

**Weaknesses:**

1. Connections with Existing Works. What's the connection and the key difference between the proposed method and existing methods? From my point of view, the proposed method looks like a combination of the decision transformer (since there is an auto-regressive formulation in Eq 10) and the decision difusser. A clearer elaboration on the connections would be great.

2. Trajectory-level return. As stated in the 2nd paragraph in the Introduction, the difussers can "generate a trajectory of states based on accumulative rewards over multiple steps as a condition". If so, why do the authors need to propose a trajectory-level return model? Is it redundant given Diffuer's existing ability to employ accumulative rewards?

3. Presentation. The whole section 3 contains too many technical details but lacks a high-level demonstration of the overall picture of the proposed method, as well as a discussion on the connections and differences with existing works.

**Questions:**

1. From my understanding, In Eq. 10, using both s_{0:t} and y(\tau) as the condition makes the formulation similar to the auto-regressive style decision transformer. Or, what's the key difference between the formulation in Eq. 10 and the decision transformer?

2. It would be great to have "a high-level demonstration of the overall picture of the proposed method, as well as a discussion on the connections and differences with existing works.", as mentioned in the Weakness.

---

> ### Author Response · Authors · 2025-11-20
> **Response to reviewer Fvh3**
>
> We sincerely appreciate your effort in reviewing our work and are grateful for your insightful comments!
>
> ---
> **Weakness 1 & Question 1: Key difference to Decision Transformer**
>
> The Decision Transformer (DT) auto-regressively generates a **single** action using $p(a_t|Rtg_{\leq t}, s_{\leq t}, a_{<t})$,  which is not modeling $p(s_{t+1:T}|s_{\leq t}, y(\tau))$ in Equation (10).
> In contrast, our CBD method, which falls under the category of diffusers, generates all future states **simultaneously** rather than using an auto-regressive manner, which is the key distinction from DT. Additionally, CBD can effectively leverage historical observations $s_{\leq t}$ to generate future states $s_{t+1:T}$, due to our augmented training process that learns to complete sequences of random lengths.
>
> A significant challenge in applying dynamic programming methods like DT to auto-bidding is its inability to handle sparse-reward settings. In DT, actions ${a}_t$ are learned through the step-wise reward signals $r_t({s}_t,{a}_t)$. However, $r_t$ could always be zero in the sparse training set. A zero reward could indicate either a poor action or a good action that did not receive a reward due to conversion randomness, making it difficult to accurately assess the value of single steps.
> Our method overcomes this limitation by conditioning on a trajectory-level cumulative reward, alleviating the need for accurate single-step rewards.
> This advantage is supported by previous works [1] and our experiments in Table 1, where our method significantly outperforms DT in sparse settings.
> We have added a detailed discussion on auto-regressive modeling to **Appendix M**.
>
> Although one could adapt DT to function as a "one-step" generator by auto-regressively generating states until the end of the trajectory, essentially implementing CBD with a decision transformer, we conducted experiments labeled **CBD-Transformer** in Table 2.
> The results of this experiment, which we have copied below from the manuscript, demonstrated poor performance of the CBD-Transformer, highlighting the importance of using a diffuser to achieve Equation (10).
>
>
> | Method | |Dense| | |Sparse| |
> | -------- | ---|--------|--- | -------- | ---| ---|
> |    | Value $\uparrow$ | ER $\downarrow$| Score $\uparrow$| Value $\uparrow$    | ER $\downarrow$   | Score $\uparrow$|
> |CBD    | 374  | 1.10  |  298  | 40.7    |  0.86      |  37.0 |
> |CBD-Transformer    |  313   | 1.35  |  210  | 31.2    |  1.21      | 21.9 |
>
>
> [1] Ajay, Anurag, et al. "Is conditional generative modeling all you need for decision-making?" ICLR 2023.
>
>
>
>
> ---
> **Weakness 2: Necessity of a trajectory-level return model**
>
> Given the model's limited capabilities, as we cannot get a perfect model, and the ever-changing preferences of advertisers, the generated trajectory might still be suboptimal, potentially causing misalignment issues. This could be analogous to advanced generative models like LLMs, which, despite their effective instruction-following abilities, still face misalignment issues.
>
> To address this during inference, we employ a trajectory-level return model as an efficient aligner to adaptively refine the final generated trajectories, ensuring they better align with advertisers' preferences.
> This approach also underscores the flexibility and explainability of using a planning-based method with diffusers.
> Together with the completer, CBD provides practical solutions for enhancing diffuser-based methods for auto-bidding in both training (by the completer) and inference (by the aligner) stages.
>
>
> ---
> **Weakness 3 & Question 2: Presentation**
>
> Figure 1 of the manuscript illustrates the inference process of CBD, and Algorithms 1&2 in Appendix D could provide an outline of the training process. We add a diagram demonstrating the high-level training procedure of CBD in **Figure 6 of Appendix D**. We sincerely hope this will effectively address any concerns you have regarding the presentation.
>
> ---
> We sincerely apologize for any misunderstandings that may have arisen and hope that our response has effectively addressed your concerns. Your feedback is extremely important to us, and we would be delighted to discuss it further with you during the rebuttal stage.

---

> ### Author Response · Authors · 2025-11-25
>
> Dear Reviewer Fvh3,
>
> We sincerely appreciate the time and effort you have dedicated to reviewing our work. As the rebuttal period will conclude in a week, we kindly ask if our response has sufficiently addressed your concerns.
>
> Briefly, we have enhanced our work by providing **clearer elaborations on existing works** like decision transformers, explaining the **benefits of a trajectory-level return model** for better alignment during the inference stage, and adding extra **illustrative figures** to improve presentation.
>
> Your feedback is extremely valuable to us, and if our responses have met your expectations, we would be deeply appreciative if you could consider increasing the score. If you have any further questions or concerns, we are more than willing to discuss them and provide any additional clarifications you may require.
>
>
> Warm regards,
>
> The Authors

---

### Meta-Review · Area_Chair_CkPL · 2026-01-06

**Summary:**

## Summary
This paper tackles the problem of auto-bidding in online advertising. The authors noticed that while diffusion models are promising for this task, they run into two main issues dynamic illegitimacy and misalignment when applied to large-scale auctions. To fix this, the paper proposes a method called CBD, aimming to mitigate the uncertainty inherent in generative models. The method is evaluated on AuctionNet and its sparse variant, showing consistent gains over baselines (e.g., +29.9% conversion value on AuctionNet-sparse). An online A/B test reports a +2.0% increase in conversions with no budget increase.



## Overall Score

Fvh3: 4 (no response)

GWa8:  4 (no response)

2xeh: 4-6 (raising score)

cGA9: 8 (keep positive score)


## Addressed concerns

* Generalization (2xeh)
* More details about the experiment results (2xeh)
* Presentation (cGA9)

## Remaining concerns

* Limited discussion of the contribution (Fvh3, GWa8)
* Further discussion on Trajectory-level return (Fvh3)
* Writing clarity (Fvh3,GWa8)
* Overstated novelty (GWa8)
* Discussion on the resolution of dynamic illegitimacy (GWa8)
* Computational cost (GWa8)

## Conclusion:

Overall, during the review phase,  two reviewers (Fvh3 and GWa8) weren't fully satisfied with writing clarity and the contribution of the paper. They also raised concerns about the limited discussion on the experiment results, computational cost and the motivation of the Trajectory-level return model. Reviewer cGA9 and 2xeh, on the other hand, leaned toward acceptance. During the discussion, the authors did a good job addressing the reviewers' questions and concerns partly, but part of the reviewers didn't follow up afterward or raise scores.
This paper has potential to be stronger with some improvements to the writing clarity, figure presentation and deeper discussion on experiments results.

**Reviewer Concerns:**

Refer to summary

**Reviewer Scores:**

Refer to summary

---

### Decision · Program_Chairs · 2026-01-26

Reject